

# An improved and continuous synchronization of the Greenland ice-core and Hulu Cave U-Th timescales using probabilistic inversion

Francesco Muschitiello[1,2]

[1]Department of Geography, University of Cambridge, Cambridge CB2 3EN, UK
[2] NORCE Norwegian Research Centre, Jahnebakken 5, 5007 Bergen, Norway

*Correspondence to*: Francesco Muschitiello (francesco.muschitiello@geog.cam.ac.uk)

**Abstract.** This study presents the first continuously measured transfer functions that quantify the age difference between the Greenland Ice-Core Chronology 2005 (GICC05) and the Hulu Cave U-Th timescale during the last glacial period. The transfer functions were estimated using an automated algorithm for Bayesian inversion that allows inferring a continuous and objective

synchronization between Greenland ice-core and Hulu Cave proxy signals. The algorithm explicitly considers prior knowledge on the maximum counting error (MCE) of GICC05, but also samples synchronization scenarios that exceed the differential dating uncertainty of the annual-layer count in ice cores, which are currently not detectable using conventional tie-point alignments or wiggle-matching techniques. The consistency and accuracy of the results were ensured by estimating two independent synchronizations: a climate synchronization based on climate proxy records, and a climate-independent

synchronization based on cosmogenic radionuclide data (i.e. $^{10}$Be and $^{14}$C). The transfer functions are up to 40% more precise than previous estimates and significantly reduce the absolute dating uncertainty of the GICC05 back to 48 kyr ago. The results highlight that the annual-layer counting error of GICC05 is not strictly correlated over extended periods of time, and that within certain Greenland Stadials the differential dating uncertainty is likely underestimated by 7.5-20%. Importantly, the analysis implies for the first time that during the Last Glacial Maximum GICC05 overcounts ice layers by 15-25% –a bias

attributable to a higher frequency of sub-annual layers due to changes in the seasonal cycle of precipitation and mode of dust deposition to the Greenland Ice Sheet. The new timescale transfer functions provide important constraints on the uncertainty surrounding the stratigraphic dating of the Greenland age-scale and enable an improved chronological integration of ice cores, U-Th-dated and radiocarbon-dated paleoclimate records on a common timeline. The transfer functions are available as supplements to this study.

## 1 Introduction

The Greenland ice-core chronology 2005 (GICC05; Rasmussen et al., 2006; Svensson et al., 2008) and the Hulu Cave speleothem U-Th timescale (Cheng et al., 2018, 2016; Southon et al., 2012) are among the most widely used independently dated chronological frameworks of the last glacial period. The timescales not only provide the backbone of some of the most



unique and detailed records of global climate change, but their robustness make them exceptionally suited for resolving the temporal structure of Dansgaard-Oeschger (DO) events and other abrupt climate shifts.

The GICC05 is based on annual-layer counting back to 60 kyr before 2000 AD (b2k) and due to the incremental nature of the counting uncertainty, it provides high internal consistency that enables accurate relative age estimates of climate events. The Greenland ice-core timescale underpins a number of high-resolution ice-core records of North Atlantic climate and

atmospheric composition. These records have become established Northern Hemisphere templates for the last glacial period and have shaped our understanding of the physical mechanisms driving rapid climate shifts (Andersen et al., 2004; Dahl-Jensen et al., 1998; Legrand and Mayewski, 1997; Schüpbach et al., 2018) and their rates of change (Jansen et al., 2020). By contrast, the Hulu Cave speleothem chronology is constructed using high-precision U-Th dating, which yields much smaller uncertainty in the absolute ages than GICC05. The U-Th timescale provides a temporal framework for the Hulu Cave

speleothem $\delta^{18}O$ data (Cheng et al., 2016; Edwards et al., 2013; Southon et al., 2012); the record constitutes a key blueprint of low-latitude hydroclimate variability, integrating intensity changes in East Asian summer monsoon and meridional shifts of the intertropical convergence zone (ITCZ; Wang et al., 2006, 2001).

Both the GICC05 and Hulu Cave U-Th timescale serve to test, improve and constrain chronologies for a wide range of paleoclimate archives and proxy records. These age scales have been used to validate and benchmark Antarctic ice-core

chronologies (e.g. Buizert et al., 2015; Sigl et al., 2016), which ultimately enable resolving the inter-hemispheric phasing of DO events (Buizert et al., 2018), and the rate of greenhouse gas emissions during the last glacial period (Bauska et al., 2021). They are also widely used to constrain paleoceanographic records with poor independent age control (Bard et al., 2013; Hughen and Heaton, 2020; Waelbroeck et al., 2019). To build a chronology for deep-sea sediment cores, proxy signals are commonly correlated to abrupt cooling and warming events observed in ice-core proxies or speleothem $\delta^{18}O$ under the assumption of

direct synchrony of climate changes. These climatically tuned chronologies, despite limiting our ability to test leads and lags between oceanic and atmospheric processes (Henry et al., 2016; Hughen and Heaton, 2020), still lay the foundations for deriving "best-guess" temporal constraints on a variety of fundamental boundary conditions of glacial ocean circulation and its coupling with the atmosphere system.

Because the Greenland ice-core and U-Th chronologies are constructed independently, the occurrence of systematic timescale

offsets and dating biases of the order of hundreds of years complicates the comparisons of events integrated in the proxy records that hinge on these timescales. Perhaps more importantly, the chronology that forms the older portion of the new IntCal20 radiocarbon calibration curve is dominantly reliant on the Hulu Cave U-Th timescale (Reimer et al., 2020). During the period spanning ~14-54 kyr b2k, a wealth of [14]C datasets have been placed on the U-Th timescale either indirectly via stratigraphic tuning of paleoclimate data to the high-resolution Hulu $\delta^{18}O$ record (Bard et al., 2013; Darfeuil et al., 2016;

Hughen and Heaton, 2020), or more directly by means of [14]C wiggle-matching (e.g. Bronk Ramsey et al., 2020; Turney et al., 2010, 2016). As a result, potential differences between the timescales –if not quantified and corrected for– can hinder a proper assessment of [14]C-dated environmental and archaeological records within the ice-core climatic framework.





Furthermore, knowledge on the existing timescale offsets is important for high-resolution studies of marine $^{14}$C (e.g. Muschitiello et al., 2019). This is crucial for those reconstructions whose chronologies are more conveniently anchored to ice-core records rather than the Hulu Cave speleothems, as is often the case for North Atlantic sediment cores that integrate common regional climatic changes (Skinner et al., 2020; Thornalley et al., 2015; Waelbroeck et al., 2019) or sites where isochronous tephra deposits can be traced between ice cores and marine records (e.g. Ezat et al., 2017; Sadatzki et al., 2019; Skinner et al., 2020, 2017). In these instances, potential discrepancies between the GICC05 and U-Th timescales can lead to an imprecise assessment of ocean $^{14}$C concentrations relative to those of the atmosphere inferred from the IntCal datasets, thus affecting the estimation of ocean carbon inventories.

In turn, resolving the differences between the GICC05 and Hulu Cave U-Th timescales can help to reduce and characterize their absolute dating uncertainty, and facilitate the comparison of ice cores and radiocarbon-dated records on a common timeline. Altogether, this is pivotal to advance our understanding of the physical mechanisms behind abrupt climate change, and to harmonize climate, environmental and archaeological records of the last glacial cycle.

There are two main types of synchronization to integrate the GICC05 and U-Th timescales: *i*) synchronization of climate records and *ii*) synchronization of cosmogenic radionuclide data. The first is based on correlation of climatic signals integrated in Greenland ice-core records and Hulu Cave $\delta^{18}$O that are assumed to be synchronous. The second is based on the correlation of externally-forced and essentially climate-independent variations in ice-core $^{10}$Be and Hulu Cave $^{14}$C records.

Despite the circularity that climate synchronizations entail, such as precluding testing the synchronicity of teleconnections between Greenland and the East Asian monsoon system, the concerns about a potentially large climate phasing between polar ice-core and Hulu Cave records during the last glacial period have been put to rest. Currently, there is ample evidence that North Atlantic and Asian monsoon climate are coupled on short atmospheric timescales (e.g. Corrick et al., 2020; Cvijanovic et al., 2013) (Fig. 1) –i.e. likely shorter than the mean resolution of the proxy records used for climate synchronization. Within dating uncertainties, ice-core data reveal bipolar synchrony during abrupt cooling and warming in Greenland, and fast global atmospheric reorganizations that propagate within a decade or less (Buizert et al., 2018; Erhardt et al., 2019; Markle et al., 2017). The teleconnection mechanism, which is likely modulated by variations in the Atlantic Meridional Overturning Circulation, is well documented and involves coherent meridional shifts of the mid-latitude storm tracks, the ITCZ, and the related monsoon systems (e.g. Pedro et al., 2018).

Since the construction of the GICC05 chronology climate synchronizations between Greenland and Hulu Cave records have been derived by identification of tie-points marking sharp transitions in both the ice-core and speleothem stratigraphies (e.g. DO events). The synchronization approach has been performed either by manual, qualitative comparison of the climate records (Svensson et al., 2008, 2006; Weninger and Jöris, 2008), or using reproducible, quantitative methods for detecting change points (Adolphi et al., 2018; Buizert et al., 2014). To present, the main methodological drawback of this approach is that it





relies on only a discrete set of stratigraphic tie-points, which prevents quantifying the alignment uncertainties in a continuous
fashion.

Synchronization of short-term variations in the production rates of cosmogenic radionuclides, such as $^{10}$Be and $^{14}$C, is another established technique to integrate the Greenland ice-core and Hulu U-Th timescales (Adolphi et al., 2018). Cosmogenic radionuclides are produced in the upper atmosphere as a result of nuclear interactions of galactic cosmic ray particles with the Earth's atmosphere (Lal and Peters, 1967). As the production rate of cosmogenic nuclides is inversely related to variations in
solar activity and the geomagnetic field, the periodic modulations integrated in ice-core $^{10}$Be and speleothem $^{14}$C records are a truly external and globally-synchronous signal, which makes these radionuclides a powerful synchronization tool. Nonetheless, synchronization using cosmogenic radionuclides presents some challenges. $^{10}$Be records in particular might be affected by archive noise and depositional/transport effects that distort the production signal integrated in ice cores. Correcting for system effects and isolating the production rate signal generally requires stacking multiple and continuous high-resolution
records, but as cosmogenic radionuclides involve very time-consuming and costly measurements, this is still a work in progress. Because the available speleothem $^{14}$C data does not always allow resolving centennial and shorter variability in atmospheric $^{14}$C production, more high-resolution $^{14}$C measurements are also urgently needed.

The approach for synchronizing cosmogenic radionuclide records involve sliding window techniques, such as cross-lagged regression (Muscheler et al., 2014a) or more commonly Bayesian wiggle-matching (BWM; Adolphi and Muscheler, 2016).
The techniques aim at matching relative changes in $^{10}$Be and $^{14}$C concentrations over a series of time windows, and by focusing on centennial-to-multi-centennial variations that are typically dominated by solar-induced –and largely periodic– changes in production rates (Vonmoos et al., 2006). Synchronization using BWM offers high precision and has proved effective during the Holocene when the offsets between the ice-core and $^{14}$C timescales are mostly systematic (Adolphi and Muscheler, 2016; Sigl et al., 2016).

However, these matching techniques depend heavily on the predefined window length (e.g. Schoenherr et al., 2019) and can lead to biased conclusions about synchrony if the timescale offsets change faster than the time window used for matching. Specifically, BWM only produces a single point of match for each window analysis, which generally spans 1,000 to 5,000 years, thus averaging out any short-term, nuanced fluctuations in the timescale difference (Adolphi et al., 2018; Adolphi and Muscheler, 2016). In addition, because of the highly autocorrelated nature of cosmogenic radionuclide records, BWM may
misrepresent the relationship between the input $^{10}$Be and $^{14}$C signals. On one hand, autocorrelation can lead to neighbouring offset estimates of the BWM to be positively correlated, which results in smoothing out the timescale transfer function. On the other hand, it may lead BWM to identify wrong correlation, thus yielding sudden –and potentially spurious– jumps in the timescale offsets (Muscheler et al., 2014a; Muschitiello et al., 2019).

With climate synchronizations standing on a limited number of stratigraphic tie-point and the latest alignment of cosmogenic
radionuclides on only five BWM estimates (Muscheler et al., 2020), a new continuous synchronization between the GICC05



and Hulu Cave U-Th timescales for the last glacial period is urgently required. The recent revision of the high-resolution Hulu Cave $\delta^{18}$O record (Cheng et al., 2016) with an updated U-Th chronology (Cheng et al., 2018), as well as improvements in the timescale precision underlying a number of ice-core $^{10}$Be records (Svensson et al., 2020), provide further motivation for re-assessing the synchronization between the two timescales. Lastly, there is a need for improved constraints during the Last

Glacial Maximum (LGM), i.e. when the timescales reach their largest offset, and assess possible fast changes in the timescale difference that are currently not detectable using wiggle-matching techniques.

In this study, these limitations are addressed by applying an automated probabilistic synchronization method to produce the first continuous transfer functions that quantify the offset between the Greenland ice-core and Hulu Cave U-Th timescales. The method minimizes the misfit between ice-core and Hulu Cave proxy records while accounting for prior knowledge on the

absolute uncertainty in annual layer identification in ice cores using a Bayesian inversion of the GICC05 maximum counting error (MCE) (Rasmussen et al., 2006; Svensson et al., 2008). The robustness of the results are ensured by determining two independent synchronizations based on climate records and cosmogenic radionuclide data, respectively. The new timescale transfer functions considerably improve the precision and accuracy of earlier estimates and strongly reduces the absolute dating uncertainty of GICC05 in the interval ~11.5-48 kyr b2k. The results also indicate large and fast fluctuations in the timescale

difference during the LGM and other cold stadial periods, suggesting previously unrecognised biases in the ice-core annual layer counting. The implications of these findings are discussed.

## 2 Data and methods

To control the accuracy of the offsets between the GICC05 and Hulu Cave U-Th timescales, two independent synchronizations were generated: a climate synchronization based on climate proxy records (CLIM), and a climate-independent synchronization

based on cosmogenic radionuclide records (COSMO). CLIM and COSMO span roughly a common interval for which both continuous high-resolution data are available.

### 2.1 Climate proxy data

The climate proxy data used in this study are presented in Fig. 2. The CLIM synchronization was established over the period ~11.5-48 kyr b2k using a combination of high-resolution $Ca^{2+}$ concentrations of mineral dust and deuterium (*d*) excess

measurements from NGRIP on the GICC05 chronology (Erhardt et al., 2019; Landais et al., 2018), and revised $\delta^{18}$O data from Hulu Cave speleothems on an improved U-Th timescale (Cheng et al., 2016).

Mineral dust aerosol in Greenland ice cores primarily originates from Asian deserts (Svensson et al., 2000). Its emissions are strongly dependent on Asian hydroclimate via concerted shifts in the latitudinal position of the ITCZ and the East Asian Summer monsoon system (Nagashima et al., 2011; Schiemann et al., 2009). While from Greenland Interstadial (GI) 13 to

Greenland Stadial (GS) 3 CLIM is constrained by DO and Heinrich (H) events expressed in the $Ca^{2+}$ record, after GS-3 the synchronization relies on *d*-excess data from NGRIP (Fig. 2a) –an indicator of water vapour source and evaporation conditions



in the North Atlantic (Masson-Delmotte et al., 2005; Pfahl and Sodemann, 2014). The use of *d*-excess data after GS-3 was preferred to $Ca^{2+}$ based on the fact that this proxy has a more clearly expressed two-phase structure during GS-2.1a-b (i.e. corresponding to HS-1; Huang et al., 2019) with trends that compare favourably to the signal recorded in Hulu Cave $\delta^{18}O$,

likely reflecting a common large-scale process (Landais et al., 2018) (Fig. 2a-b). It should be noted that the proxy signals in $Ca^{2+}$ and *d*-excess are qualitatively similar, that the relative phasing between the aerosol and water isotope records are smaller than the CLIM synchronization uncertainty (Erhardt et al., 2019), and that the CLIM results after GS-3 are mainly independent of the proxy parameter used. Because both NGRIP $Ca^{2+}$ and *d*-excess indirectly register lower-latitude hydroclimate changes mediated by latitudinal migrations of the ITCZ, they are suitable for direct comparison to Hulu Cave $\delta^{18}O$ data, which integrate

ITCZ-related shifts in monsoon rainfall over East Asia (Wang et al., 2006, 2001) with comparable durations (Fig. 2d).

As for Hulu Cave, the climate record used here is a recently augmented data set that incorporates higher temporal resolution $\delta^{18}O$ measurements (Cheng et al., 2016) underpinned by an age model that builds upon an extended and revised set of U-Th dates (Cheng et al., 2018) (Fig. 2b-c). These updates greatly helped to improve the chronological precision and accuracy of the Hulu Cave $\delta^{18}O$ record.

**2.2 Cosmogenic radionuclide data and model-based $\Delta^{14}C$ reconstruction**

The cosmogenic isotope production data used in this study are shown in Fig. 3. The COSMO synchronization was established using independent $^{10}Be$ records from Greenland and Antarctic ice cores placed on the GICC05 timescale, and U-Th dated $^{14}C$ measurements from Hulu Cave speleothems. As discussed below, because the sampling frequency of the Hulu Cave $^{14}C$ measurements is not always sufficiently fine to resolve any structure in production rates necessary for aligning the ice-core

and speleothem data, direct synchronization estimates were only obtained for the intervals ~11.5-26.5 and ~36.5-45 kyr b2k.

For the purpose of this work, only published and publicly available $^{10}Be$ records were considered. The compilation is made of eight individual $^{10}Be$ data sets including GRIP (Adolphi et al., 2014; Baumgartner et al., 1997; Wagner et al., 2001; Yiou et al., 1997), NEEM (Zheng et al., 2021), GISP2 (Finkel and Nishiizumi, 1997), NGRIP, EDC, EDML and Vostok (all presented in Raisbeck et al., 2017), and WDC (Muschitiello et al., 2019) (Fig. 3a). The Greenlandic records are directly standing on the

GICC05, whereas the Antarctic records are robustly tied to it using high-precision synchronization of volcanic and/or solar tracers. The records from EDC and EDML ice cores were initially placed on the WD2014 chronology (Buizert et al., 2014; Sigl et al., 2016) using common volcanic markers (Buizert et al., 2018). These records and the WDC data set were subsequently transferred from WD2014 onto the GICC05 using a new bipolar synchronization over the interval ~12-60 kyr b2k based on densely-spaced volcanic tie points (Svensson et al., 2020) and assuming linear interpolation between the ties. The Vostok

record instead was stratigraphically linked to GICC05 using peaks and troughs in the $^{10}Be$ profile and volcanic markers (Raisbeck et al., 2017). The $^{10}Be$ data compilation presented here is comparable to that used in Adolphi et al. (2018). However, it incorporates a new $^{10}Be$ record from WDC spanning the last deglaciation, new data from NEEM covering the entirety of the last glacial period, and generally offers an increased dating precision of all the ice-core records from Antarctica.



The [10]Be concentrations of the eight cores were converted to fluxes using accumulation rates calculated from annual layer
thicknesses adjusted for ice thinning. For the Greenland ice cores, the layer thicknesses were derived from the GICC05
timescale and corrected using cumulative strain estimates based on ice-flow modelling (Dahl-Jensen et al., 1993; Seierstad et
al., 2014). Similarly, WDC accumulation rates over deglaciation were reconstructed by correcting  the annual layer thickness
from the WD2014 timescale for flow-induced layer thinning (Fudge et al., 2016). For the remaining Antarctic ice cores, the
layer thicknesses and strain rates were derived from the AICC2012 timescale (Veres et al., 2013).

The [10]Be flux records were then combined using a Monte Carlo approach. The individual records were first normalized by
dividing each value by the record mean. For each Monte Carlo realization, every normalized [10]Be record was randomly
resampled within its analytical uncertainty (assuming an error value stemming from a Gaussian distribution) and interpolated
to a common time step of 5 years. The resampled records were stacked together by computing a weighted average based on
the mean sampling resolution of the original records contributing to the stack. This procedure was repeated to construct 10,000
realizations of the [10]Be stack and establish its mean and standard deviation (Fig. 3a).

As non-production related processes may affect the [10]Be production rate signal integrated in ice cores, it is becoming common
to apply a climate correction on the individual records to minimize climate-related transport and depositional effects on [10]Be
fluxes (Zheng et al., 2021). However, it is still unclear as to what extent this approach can effectively remove the potential
climate signal in [10]Be data, and previous studies have shown that the synchronization outcomes are generally unaffected by
climate corrections (Adolphi et al., 2018; Adolphi and Muscheler, 2016). For these reasons and because stacking multiple
records is per se an equally efficient method to reduce the noise and isolate the production rate signal (Zheng et al., 2020), a
climate correction was not applied here. Furthermore, it should be noted that the [10]Be flux data was low-pass filtered before
synchronization (see details below), which also helps filtering out climate-driven high-frequency fluctuations and noise
(Heikkilä et al., 2014).

The radiocarbon data used for this study consist of a combination of earlier and recent high-resolution [14]C measurements from
three Hulu Cave speleothems (MSD and MSL, Cheng et al., 2018; H82, Southon et al., 2012) (Fig. 3c). The Hulu Cave data
set stretches back to ~54 kyr b2k and comprises ~650 [14]C determinations paired with U-Th dates, which allows anchoring the
radiocarbon record to a precise and absolute timescale. To estimate atmospheric [14]C variations through time, the radiocarbon
ages were converted to absolute [14]C concentrations (expressed as $\Delta$[14]C) via correction for decay and fractionation relative to a
standard (Stuiver and Polach, 1977). Because the $\Delta$[14]C signal in speleothems is diluted by other $CO_2$ sources, which have $\Delta$[14]C
signatures different than that of the atmosphere, such as soil gas, soil organic matter, and [14]C-devoid limestone above the cave,
the Hulu Cave [14]C record needs to be corrected for the dead carbon fraction (DCF). Based on estimates from speleothem H82,
a constant DCF of 450 ±70 [14]C years was applied for the entire record. It should be noted that a few sections of the Hulu Cave
[14]C record are still not sufficiently resolved to integrate multi-centennial and shorter variability in atmospheric [14]C production
(Fig. 3c-d), and were therefore not considered for synchronization. For this reason COSMO only covers two time intervals





characterized by higher-resolved and more evenly sampled data, i.e. 11.5-26.5 and 36.5-45 kyr b2k, which provide the most reliable synchronization estimates.

A more direct atmospheric $^{14}$C reconstruction for the last glacial period is provided by the Lake Suigetsu record, which has been recently matched to the new extended Hulu Cave data (Bronk Ramsey et al., 2020). However, even though Lake Suigetsu
offers an unfiltered atmospheric data set with high temporal resolution, it is still unclear as to whether the high-frequency $\Delta^{14}$C variations observed in the Suigetsu record are a genuine cosmogenic production signal or noise in the sedimentary environment (Bronk Ramsey et al., 2020). For this reason, this data set was not considered here and will be the subject of a future investigation.

To relate ice-core $^{10}$Be fluxes to atmospheric $^{14}$C production rates reconstructed from Hulu Cave, carbon cycle effects have to
be taken into account using appropriate modelling. The approach to modelling the carbon cycle used here follows the methodology outlined in Adolphi et al. (2018). In short, to generate a $^{10}$Be-based $\Delta^{14}$C record, each realization of the $^{10}$Be flux stack was converted to $\Delta^{14}$C variations using a box-diffusion carbon cycle model (Siegenthaler et al., 1980) (Fig. 3b). The model was run under the assumption that $^{10}$Be and $^{14}$C production rate changes are directly proportional, i.e. scaling up $^{10}$Be by 30% (Herbst et al., 2017), and prescribing boundary conditions in which ocean ventilation is 75% of its pre-industrial value.
There are considerable uncertainties surrounding the state and long-term evolution of the carbon cycle, which hinders a straightforward comparison between modelled and reconstructed $\Delta^{14}$C (Köhler et al., 2006). However, the relative variations in $^{10}$Be-based $\Delta^{14}$C are largely independent of the variable state of the carbon cycle and the phase of centennial $\Delta^{14}$C anomalies are not affected by carbon cycle perturbations (Adolphi and Muscheler, 2016). Given that the phase is the property of the signal most relevant for the synchronization procedure, to minimize systematic and transient carbon cycle effects on $\Delta^{14}$C
levels, COSMO was constrained by using primarily filtered $\Delta^{14}$C anomalies.

To more directly compare centennial changes in modelled and reconstructed $\Delta^{14}$C anomalies, prior to synchronization the high-resolution data were band-pass filtered to isolate variations between 150 and 500 years. Because of the sinusoidal nature of cosmogenic radionuclide production rates, these band-passed high-frequency anomalies in the input and target timeseries yield a certain degree of autocorrelation, which, if not accounted for, can lead to alignments that are out-of-phase. Thus, to arrive to
a reliable synchronization, sufficient long-term structure to guide the alignment of the high-frequency $\Delta^{14}$C anomalies is desirable. This problem was tackled by constraining the COSMO synchronization with a multi-parameter approach that allows aligning multiple inputs and targets simultaneously (see sections 2.3.1 and 2.3.4). As such, a second set of input and target timeseries was employed using high-passed or unfiltered versions of the $\Delta^{14}$C records that isolate longer multi-millennial variations in production rates. These complementary data of low-frequency $\Delta^{14}$C anomalies span the intervals 18-26.5 and 40-
44 kyr b2k, which have been previously exploited to infer millennial-scale covariability in measured and modelled $\Delta^{14}$C records (Adolphi et al., 2018). The most significant common low-frequency structures present in the younger window –and likely driven by a series of solar minima– are the centennial-scale fluctuations at ~19, 22 and 24 kyr b2k, whereas the older window is characterized by the well-defined geomagnetic excursion associated with the Laschamp event at ~42 kyr b2k (Fig.



3b-c). Over the interval 18-26.5 kyr b2k, the complementary records were high-pass filtered to highlight variations on
timescales longer than 5000 years, which has an effect similar to linearly detrending the data. For the 40-44 kyr b2k interval,
the records were simply demeaned in order to preserve the large production anomaly that defines the Laschamp event.

Given that these longer-term $\Delta^{14}$C anomalies are potentially prone to uncertainties in carbon cycle changes, the uncertainty
estimates of the $^{10}$Be-based $\Delta^{14}$C records were inflated accordingly. The inflation was based on conservative estimates of the
uncertainties that accompany $\Delta^{14}$C modelling experiments during the last glacial period (Köhler et al., 2006). Following the
rationale by Adolphi et al. (2018), the uncertainty of the modelled $\Delta^{14}$C data was increased by ±30‰ (1σ) during the interval
18-26.5 kyr b2k  –a period that was possibly not affected by any major carbon-cycle change (Bauska et al., 2021; Eggleston
et al., 2016). By contrast, a larger uncertainty of ±50‰ was assigned during the interval 40-44 kyr b2k, when the effects of
changing carbon cycle were likely more significant. A more comprehensive discussion on all the major aspects of the $^{10}$Be
flux conversion into $\Delta^{14}$C, propagation of errors, and the uncertainty associated with carbon-cycle modelling can be found in
Adolphi et al. (2018) and Adolphi and Muscheler (2016).

### 2.3.1 Probabilistic algorithm for proxy-data synchronization

As discussed in Section 1, tie-point and wiggle-matching synchronizations have inherent problems that limit estimating the
alignment of proxy timeseries in a truly continuous fashion. The alignment uncertainty of tie-point synchronizations is poorly
characterized, tie points can be difficult to reproduce, and even when they are defined statistically the synchronization still
does not account for potential shared signal structures in between consecutive ties. Wiggle-matching techniques on the other
hand offer high-precision synchronizations. However, they often require averaging over long time windows, which smooth
out information on the short-term relationship between proxy timeseries. This limitation becomes particularly significant when
the offsets between the timescales to be synchronized change faster than the span of the time window used for matching.

Probabilistic alignment methods have a unique and underexploited potential to correlate proxy timeseries and move away from
point-wise and wiggle-matching synchronization techniques. They are especially well suited for establishing continuous
alignments and can help matching previously untapped common structures in the signal of climate and cosmogenic
radionuclide records. These methods are fully automated and have the advantage of ensuring reproducibility, deriving credible
bands associated with the alignment process, and inferring the probability of synchronization solutions based on prior
constraints on accumulation rates (e.g. Lin et al., 2014; Muschitiello et al., 2020; Parrenin et al., 2015).

In this study, a continuous synchronization of the GICC05 to the Hulu Cave U-Th timescale is established using an appositely
developed algorithm for probabilistic inversion. The inverse problem is formulated using a Bayesian framework in order to
sample the full range of possible GICC05-Hulu Cave synchronization scenarios and explicitly build in prior ice-core
chronological constraints. The inverse scheme is linked to a forward model, which –assuming that the U-Th timescale is
absolute– simulates the age offset history between GICC05 and Hulu Cave in response to changes in Greenland ice





accumulation over time. The link requires a likelihood function, which quantifies how probable the alignment between ice-core and speleothem records is given a particular simulated ice-core depositional history.

The numerical approach builds upon previous work using a hidden Markov model for automated synchronization of paleoclimate records (Muschitiello et al., 2020, 2019; Sessford et al., 2019; West et al., 2019). The model employed here uses constraints imposed by the MCE, i.e. the accumulated absolute annual layer counting error of the Greenland ice-core chronology, to deform the entirety of an input timeseries (on the GICC05 timescale) onto a target (on the Hulu Cave U-Th timescale). The method weighs the probability of any given alignment based on the misfit between the input and the target, and eventually 'finds' a sample of alignments between Greenland ice cores and Hulu Cave speleothems that are physically coherent with the absolute dating uncertainty of GICC05 and some of its counting error properties. The method deployed here is adaptable to a variety of formulations of the inverse problem and to using multiple input and target records simultaneously when determining the alignment –an implementation that considerably improves the robustness of the synchronization estimates.

**2.3.2 Inverse modelling approach**

The inverse approach to modelling a continuous synchronization between Greenland ice cores and Hulu Cave speleothem records requires a forward model function $S$ that relates the model and data space and can be defined as follows:

$$A^{sync} = S(X) \tag{1}$$

where $A^{sync}$ represents the simulated aligned data and $X$ the synchronization parameters that produce $A^{sync}$ values. The simulated data in the optimization problem are described by random variables of age $A_i^{Hulu}$, which relate the GICC05 age of the $i^{th}$ data point in the input ice-core record (with $i = 1, 2, \ldots, n$) for all $n$ input data points to unknown U-Th ages on the target Hulu Cave record. One can thus express the synchronization vector $A^{Hulu} = (A_1^{Hulu}, A_2^{Hulu}, \ldots, A_n^{Hulu})$ as a series of monotonically increasing assignments of U-Th ages for every measurement in the input record with GICC05 age $A^{GICC05} = (A_1^{GICC05}, A_2^{GICC05}, \ldots, A_n^{GICC05})$. Hence, given the input data, the model learns which values of $X$ simulate ice-core data that most closely match the chosen target Hulu Cave record.

Since the synchronization process is fundamentally uncertain, we apply probability theory in the age assignment of the vector $A^{Hulu}$. To infer the uncertainty on the optimal model, a likelihood cost function is required. This is the probability for a given residual misfit between the aligned ice-core data and the target speleothem record, given a model with a particular set of parameter values. In the synchronization problem posed here, the likelihood function determines the goodness of the fit between the input and the target by weighing the competing needs of a high $R^2$ coefficient of determination and a small root-mean-square deviation (RMSD). This metric is expressed as follows:

$$M(X) = \left(1 - \sum_{i=1}^{n} \frac{|D_i^{Hulu} - D_i^{ice}|}{|D_i^{Hulu} - \overline{D}^{Hulu}|}\right) + \frac{1}{n}\sum_{i=1}^{n} \frac{|D_i^{Hulu} - D_i^{ice}|}{\sigma_i} \tag{2}$$





where the likelihood function is defined as:

$$L(\mathbf{X}) = e^{-M(\mathbf{X})} \tag{3}$$

The first argument in Eq.2 represents $1 - R^2$ and the second RMSD, $n$ is the number of synchronized data points in the input ice-core record, $D_i^{ice}$ and $\sigma_i$ are the proxy value for the $i^{\text{th}}$ point in the input for a proposed alignment age $A_i^{Hulu}$ and its uncertainty, respectively, $D_i^{Hulu}$ is the corresponding proxy value in the target on its original U-Th timescale, and $\overline{D}^{Hulu}$ is the

mean value of the target record. In Bayesian statistical terms, the likelihood can be written as $P(D^{ice}|A^{Hulu}, X)$, which defines the probability that the input record $D^{ice}$ would be observed for a particular alignment $A^{Hulu}$. The Bayes' rule is then applied to incorporate in the model glaciological information on unobserved changes in Greenland ice accumulation. The rule combines the likelihood probability with the prior probability on the unknown model parameters and is written as follows:

$$P(A^{Hulu}, X|D^{ice}) \propto P(D^{ice}|A^{Hulu}, X) P(A^{Hulu}|X) \tag{4}$$

The prior probability distribution is defined by $P(A^{Hulu}|X)$, which expresses the probability that a given alignment $A^{Hulu}$ would be observed independent of any $D^{ice}$ data (i.e. the probability of a given sequence of U-Th age assignments to the data points of the input ice-core record), and that is here referred to as the "synchronization model". The rule is ultimately applied to compute the posterior probability for any given alignment $P(A^{Hulu}, X|D^{ice})$, which expresses the relationship between the data and the model space, and is proportional to the product of likelihood and prior.

**2.3.3 Forward model**

The synchronization model estimates the probability of a given alignment that relates GICC05 and U-Th ages independent of the input ice-core data. To avoid sampling outside a physically reasonable range and to identify a sample of optimal synchronizations, this probability is based on prior knowledge that time-dependent changes in the offset between the GICC05 and Hulu Cave timescales are largely limited by the constraints imposed by the absolute counting error of GICC05. To achieve

this, the Bayesian formulation of our inversion approach requires a numerical forward synchronization model that simulates feasible alignments between ice-core and Hulu Cave records. The forward simulation is here defined by a simple mathematical representation of the GICC05-Hulu Cave age relationship that effectively allows linearly stretching and compressing the ice-core chronology relative to the U-Th timescale.

The synchronization is represented at any location $i$ on the GICC05 timescale by a piecewise linear function on a time interval

$[t_0, t_n]$. The function is described by a sequence of segments $\phi_i$ connected by strictly monotonic points $(A_i^{GICC05}, A_i^{Hulu})$ and $(A_{i+1}^{GICC05}, A_{i+1}^{Hulu})$, and can be written as:

$$\phi(A^{GICC05}) = A_i^{Hulu} + \frac{A_{i+1}^{Hulu} - A_i^{Hulu}}{A_{i+1}^{GICC05} - A_i^{GICC05}} (A^{GICC05} - A_i^{GICC05}), \tag{5}$$





$$A^{GICC05} \in \left[ A_i^{GICC05}, A_{i+1}^{GICC05} \right], \qquad i = 1, \dots, n-1$$

where $n \geq 2$, $t_0 = A_1^{GICC05} < \cdots < A_n^{GICC05} = t_n$. This formulation imposes synchronization paths that ensure that the vector

$\boldsymbol{A_{Hulu}}$ increases monotonically with the age of the input record $\boldsymbol{A^{GICC05}}$. To avoid sudden jumps between the segments $\phi_i$ –

and by extension in the timescale offset– the continuity of $\phi\left(A^{GICC05}\right)$ is checked by factoring in the following property:

$$\lim_{A^{GICC05}\to a^-} \phi\left(A^{GICC05}\right) = \phi(a) = \lim_{A^{GICC05}\to a^+} \phi\left(A^{GICC05}\right) \tag{6}$$

that is if one approaches a given $A^{GICC05} = a$ from either the left or the right, $\phi\left(A^{GICC05}\right)$ will approach $\phi(a)$. The continuous

piecewise linear function adopted here is composed by a sequence of four segments such that:

$$\phi\left(A^{GICC05}\right) = \begin{cases} \phi_1\left(A^{GICC05}\right), & A^{GICC05} \in [A_1'^{GICC05} = t_0, A_2'^{GICC05}] \\ \cdots \\ \phi_4\left(A^{GICC05}\right), & A^{GICC05} \in [A_4'^{GICC05}, A_5'^{GICC05} = t_n] \end{cases} \tag{7}$$

where $A'^{GICC05}$ consist of ascending GICC05 ages corresponding to the vertices of the piecewise linear functions $\phi\left(A^{GICC05}\right)$.

The segments $\phi_1$ and $\phi_4$ have equal slopes defined as $s_1 = s_4 = \frac{A_5'^{Hulu} - A_1'^{Hulu}}{t_n - t_0}$, whereas $\phi_2$ and $\phi_3$ have respectively slopes

$s_2 = \frac{A_3'^{Hulu} - A_2'^{Hulu}}{A_3'^{GICC05} - A_2'^{GICC05}}$ and $s_3 = \frac{A_4'^{Hulu} - A_3'^{Hulu}}{A_4'^{GICC05} - A_3'^{GICC05}}$ where $A'^{Hulu}$ denote the U-Th age assignments for the vertices $A'^{GICC05}$. The

slopes of the segments reflect the rate at which the age relationship between the GICC05 and Hulu Cave U-Th timescales

changes over two adjacent vertices, whereby $s > 1$ implies a linear stretching of GICC05 and $s < 1$ implies a compression.

### 2.3.4 Model parameters and priors set up

The main variables that characterize the age relationship between the two timescales are the U-Th age assignments for the start

and end data points of the input record ($A_1'^{Hulu}$ and $A_5'^{Hulu}$), the GICC05 ages corresponding to the vertices of the segments

$\phi_2$ and $\phi_3$ ($A_2'^{GICC05}$, $A_3'^{GICC05}$, and $A_4'^{GICC05}$), and the slopes of the segments $\phi_2$ and $\phi_3$ ($s_2$ and $s_3$).

The first and last U-Th age assignments of the input ice-core data are given by:

$$A_1'^{Hulu} = t_0 + \Delta T_0 \tag{8}$$

and

$$A_5'^{Hulu} = t_n + \Delta T_n \tag{9}$$

where $\Delta T_0$ and $\Delta T_n$ are treated as model parameters that define the timescale offset $A^{Hulu} - A^{GICC05}$ at $t_0$ and $t_n$, respectively.

The age difference between the GICC05 and U-Th timescales at $t_0$ (11.5 kyr b2k for both CLIM and COSMO) and $t_n$ (48 kyr

b2k for CLIM and 45 kyr b2k for COSMO) is relatively well constrained (Adolphi et al., 2018; Muscheler et al., 2020), and

conservative priors based on the published estimates have been assigned using uniform probability distributions expressed in

years. For CLIM, the following priors were used: $P(\Delta T_0) = U(-50, 50)$ and $P(\Delta T_n) = U(-1000, 1000)$. Since COSMO




was divided into two synchronization sub-problems, the priors were assigned as follows: $P(\Delta T_0) = U(-50, 50)$ and $P(\Delta T_n) = U(-750, 750)$ for the interval 11.5-26.5 kyr b2k, and $P(\Delta T_0) = U(-750, 750)$ and $P(\Delta T_n) = U(-1000, 1000)$ for the interval 36.5-45 kyr b2k.

The parameters $A_2'^{GICC05}$, $A_3'^{GICC05}$, and $A_4'^{GICC05}$ were assigned a uniform probability $P(A_{2,3,4}'^{GICC05}) = U(t_0, t_n)$ such that $t_0 < A_2'^{GICC05} < A_2'^{GICC05} < A_2'^{GICC05} < t_n$. Upon these depend the slopes $s_2$ and $s_3$, which can be expressed as:

$$s_2 = 1 + \frac{\tau_a \cdot k_a}{\Delta A_{23}'^{GICC05}} \tag{10}$$

and

$$s_3 = 1 + \frac{\tau_b \cdot k_b}{\Delta A_{34}'^{GICC05}} \tag{11}$$

where $\Delta A_{xy}'^{GICC05}$ reflects the difference between the GICC05 age of the vertices of the segments $\phi_x$ and $\phi_y$, $\tau$ indicates the differential MCE of GICC05 associated with $\Delta A_{xy}'^{GICC05}$ ($|MCE_x - MCE_y|$), and $k$ is a scaling parameter that controls the rate at which $\tau$ is allowed to change over the time interval $\Delta A_{xy}'^{GICC05}$. In other words, $k_a$ and $k_b$ regulates how much GICC05 may be linearly stretched or compressed for a given value of the maximum relative counting error (i.e. the rate of change of the MCE). Since one of the objectives of this study is to invert changes in the GICC05-Hulu Cave U-Th timescale difference brought about by possible unaccounted biases in the ice-core annual layer counting, the prior knowledge should not be too restrictive and allow sampling relative counting errors greater than the nominal values imposed by the GICC05, which during the last glacial period range ~5-10% (Svensson et al., 2008, 2006). Therefore, the scaling parameters $k_a$ and $k_b$ were assigned prior distributions that allow the model exploring relative counting errors up to 10 times greater by adopting a normal probability $P(k_{a,b}) = N(0, 10^2)$. While this approach may result in sampling synchronization paths that violate the absolute counting error of GICC05, the model is still constrained to an effectively realistic range of MCE values. This was done by balancing Eq.2 with an appropriate term that overweighs the cost of the synchronization such that the condition $\Delta T_j < 1.75 \cdot MCE_i$ (with $j = 1, 2, ..., n-1$) is always met. This implementation allows considering GICC05-Hulu Cave synchronizations that marginally exceed the range allowed by the MCE (as is generally the case for the Holocene, e.g. Adolphi and Muscheler, 2016), as well as accommodating propagation of the age errors associated with the U-Th timescale (Fig. 3d). It should be noted that the forward model and the prior information used in this study provide by no means a realistic representation of the complexity that characterizes the ice-core layer counting procedure and its uncertainty (Rasmussen et al., 2006; Svensson et al., 2006). Rather, the approach deployed here aims at inferring first-order estimates of the synchronization uncertainty, while approximating the layer-counting structure of the GICC05 timescale.

### 2.3.5 Determining the posterior distribution





Because of the nonlinear nature of the synchronization problem and the fact that there are far too many alignments to calculate all their probabilities, a stochastic Monte Carlo method is required to explore the posterior distribution in a computationally efficient way. Calculation of the posterior probability proceeds by sampling an initial value for each unknown model parameter
from the associated prior distributions using Markov chain Monte Carlo (MCMC), which is here driven by a random walk Metropolis sampler (Vihola, 2012). The sampler perturbs the current model $\boldsymbol{X}^t$ at the state $t$ by a random walk in the parameter space and obtain a candidate model $\boldsymbol{X}^{t+1}$. Using the Metropolis algorithm (Metropolis et al., 1953) the candidate model is accepted as the next state of the chain with a probability given by:

$$P_{X_i^t \to X_i^{t+1}} = \min\left(\frac{L(\boldsymbol{X}_{t+1})}{L(\boldsymbol{X}_t)}, 1\right)$$

(12)

The input and target measurement errors were assumed to be uncorrelated and normally distributed with zero mean and were randomly propagated in each MCMC iteration (note that for COSMO the error propagation includes the DCF uncertainty). Sampling was repeated for $10^6$ MCMC iterations after disregarding a burn-in time of $10^5$ steps to ensure the statistical independence of the model parameters. The sample from the remaining iterations was used to estimate the time-dependent posterior median GICC05-Hulu Cave synchronization and its credibility bands. This was deemed to be a sufficiently long
MCMC run for the simulation to reach convergence, as further assessed by initializing the sampler at different values and visually comparing the resulting synchronization estimates.

To achieve more effective mixing of the model parameters and deal with multi-modal posterior probability distributions, the likelihood function was tempered and multiple random restarts were used. Tempering flattens the posterior distribution, which in turn allows easier movement of the sampler between individual modes (e.g. Marinari and Parisi, 1992). Random restarts
allow the algorithm running for a given number of iterations and reset the likelihood cost. For each restart, the best fit obtained in the preceding scoring phase is set as the starting point for a new Markov chain. This facilitates the algorithm exploring the existence of multiple local maxima for the likelihood function that may be otherwise overlooked using a single long Markov chain. For the present application 10 random restarts were used and sensitivity tests indicate that 3 restarts are already sufficient for both CLIM and COSMO to reach convergence of the simulated chain to the target posterior distribution.

**3 Results and discussion**

**3.1 Synchronization and timescale transfer functions**

The inverted CLIM and COSMO synchronization histories and related GICC05-Hulu Cave timescale transfer functions are presented in Figures 4 and 5, respectively. Assuming that the U-Th timescale is accurate, it can be observed that throughout the last glacial period both transfer functions are well within the MCE limits of GICC05 and reveal essentially coherent
timescale offset histories ($\Delta T = A^{Hulu} - A^{GICC05}$).





Both CLIM and COSMO indicate that at the onset of the Holocene GICC05 is slightly older than the U-Th timescale, yielding respectively a $\Delta T$ of $-15^{-2}_{-30}$ years (95% credible range) and $-40^{-20}_{-45}$ years (Figs. 4-5). Between ~11.5 and 16 kyr b2k, GICC05 is gradually stretched and becomes younger than the U-Th timescale, resulting in $\Delta T$ values of $+60^{+90}_{-20}$ years (CLIM) and $+145^{+230}_{+50}$ years (COSMO). In the interval ~16-22 kyr b2k, corresponding to the early stage of GS-2, GICC05 is further

stretched but slightly faster than allowed by the annual counting error, reaching a maximum $\Delta T$ of $+485^{+650}_{+290}$ years (CLIM) and $+480^{+590}_{+270}$ years (COSMO), which is as large or a little larger than the MCE permits. From ~22 to 26.5 kyr b2k, which corresponds to GS-3 and approximately in phase with the global LGM ice-volume peak (Hughes and Gibbard, 2015), again the GICC05 annual-layer count changes faster than the counting error allows, highlighting a quick compression of the timescale with $\Delta T$ values dropping to $+95^{+155}_{-85}$ years (CLIM) and $+120^{+260}_{-90}$ years (COSMO). Between ~26.5 and 36.5 kyr b2k, only

estimates from CLIM are available and suggest that the compression of GICC05 likely continues until ~27 kyr b2k, when $\Delta T$ exhibits a minimum value of $-40^{+20}_{-160}$ years. During the interval ~26.5-29 kyr b2k, which covers the onset of GS-3, GI-3 and GS-4, CLIM reveals another stretch of GICC05 that is relatively faster than that expected by the layer-counting error, yielding a $\Delta T$ as high as $+265^{+325}_{+185}$ years. Between ~29 and 38 kyr b2k, that is the period spanning from GS-5 to GI-8, GICC05 is gradually compressed until reaching a minimum $\Delta T$ value of $0^{+130}_{-80}$ (CLIM) at ~37.8 kyr b2k. The interval ~38-40 kyr b2k,

which corresponds to GS-9, is marked once again by a rapid stretch of the GICC05 timescale beyond the bounds permitted by the MCE, resulting in maximum $\Delta T$ values of $+340^{+455}_{+270}$ years (CLIM) and $+320^{+545}_{+155}$ years (COSMO). Before 40 kyr b2k, both CLIM and COSMO show a stabilization of the timescale offsets at a $\Delta T$ of ~300 years.

The largest $\Delta T$ excursions between ~16 and 26.5 kyr b2k are overall robust and independently checked (Figs. 4-5). In CLIM, these are primarily driven by the alignment between the GS-3 dust peaks in the Greenland ice cores (Rasmussen et al., 2008)

and the well-defined declines in East Asian monsoon recorded in the Hulu $\delta^{18}$O data (Fig. 4a). COSMO on the other hand hinges on a good peak-to-peak fit of the high-frequency $\Delta^{14}$C anomalies throughout the interval ~11-22 kyr b2k, which is further supported by a coherent match of their relative amplitudes (Fig. 5a). Between ~18 and 25 kyr b2k, the alignment of the high-frequency anomalies is possibly less robust but the overall synchronization is consistent and validated by the match of the low-frequency $\Delta^{14}$C structures (Fig. 5b).

Notably, during GS-2 the transfer function error associated with CLIM is substantially larger than that inferred from COSMO (Fig. 4c, 5c). This is primarily due to the low signal-to-noise ratio in the climate records across the Lateglacial and LGM, where the lack of any clear shared structures in the input and target data results in the climate-based alignment to be relatively more uncertain. This, together with the broad priors for the relative counting error used to invert the synchronization (see section 2.3.4), likely combines to make the uncertainties around ~22 kyr b2k an overestimation of the real error that

accompanies the $\Delta T$ estimates from CLIM.

Before ~ 26.5 kyr b2k, the $\Delta T$ results indicate two fast stretches of the GICC05 timescale, i.e. one inferred from CLIM at ~27-29 kyr b2k, and another inferred from both CLIM and COSMO at ~38-40 kyr b2k (Figs. 4-5). The first stretch is well



constrained by the match of GI-3, GS-4 and GI-4 in NGRIP $Ca^{2+}$ and the corresponding monsoon events integrated in the Hulu Cave $\delta^{18}O$ record (Fig. 4a). The second one takes place during GS-9. Analogously, this stretch of the GICC05 is supported by

a robust correlation of the climate signals straddling GI-8 and GI-9 (Fig. 4a), as well as by a consistent peak-to-peak fit of the high-frequency $\Delta^{14}C$ anomalies throughout ~36.5-40 kyr b2k (Fig. 5a). Across the Laschamp event at 40-44 kyr b2k, the COSMO transfer function is robustly constrained via the simultaneous synchronization of the low-passed and the demeaned cosmogenic radionuclide data (Fig. 5a-b). Before 44 kyr b2k, COSMO relies solely on the low-passed records and because the synchronization depends on the exact phasing of their high-frequency modulations, the match is here more tentative and the

$\Delta T$ estimates are thus surrounded by larger uncertainties (Fig. 5c).

A comparison of the two independent timescale transfer functions alongside published $\Delta T$ estimates is presented in Figure 6. Because of the low sampling resolution between 26.5 and 36.5 kyr b2k in the Hulu Cave $^{14}C$ data, direct estimates of the timescale difference using cosmogenic radionuclide data were not attained over this interval. To fill the gap in the COSMO synchronization, an interpolation model was fitted to the transfer function results using the approach described in Adolphi et

al. (2018). The interpolation is based on an autoregressive (AR) process, which takes into account some of the autocorrelation structure imposed by the GICC05 timescale. However, the AR model used here ($\Phi = 0.975, \sigma = 1$) provides a more conservative estimate of the interpolation uncertainty of the transfer function, as it allows the model exploring a relative counting error domain that reflects the mean layer miscount rate inferred from the inverse analysis associated with COSMO (Fig. 6), which is ~3 times faster than that allowed by the MCE constraints.

Comparing the inferred transfer functions highlights that CLIM and COMSO are statistically indistinguishable throughout the last glacial period. More importantly, the timescale offset histories are in good agreement with independent $\Delta T$ estimates based on BWM of cosmogenic radionuclide records and match points between GICC05 and the $^{14}C$ timescale. The uncertainty bounds of the new transfer functions are overall ~40% and ~25% narrower (for CLIM and COMSO respectively) than previous estimates based on BWM (Adolphi et al., 2018; Muscheler et al., 2020). Uncertainties are on average ~10% smaller during

deglaciation and LGM (i.e. after 26 kyr b2k), whereas the precision is improved by up to ~60% during Marine Isotope Stage 3 (i.e. prior to 26 kyr b2k).

Not only the new timescale transfer functions are considerably more precise, but the continuous synchronization approach applied here brings to light a more complex GICC05-Hulu Cave $\Delta T$ history than previously assumed. Moreover, the relaxed priors of the synchronization model allowed identifying a number of potential fast changes in the timescale difference –i.e.

features that would have gone undetected had the model been more tightly constrained by the nominal relative counting error of GICC05.

**3.2 Differential dating uncertainty of GICC05**

Assuming that the U-Th timescale is absolute, a new picture emerges showing that the identification of uncertain annual layers in the GICC05 is potentially less accurate than previously thought (Fig. 6-7). The GICC05 timescale appears to be either



missing or gaining time beyond its relative counting error during a number of stadials (Fig. 7c). Too few annual layers have been identified within GS-1, H1/GS-2, GS-4 and H4/GS-9, whereas too many layers have been counted over LGM/GS-3. This in principles challenges the layer counting method and uncertainty estimates and implies that the bias in the GICC05 layer counting is not systematically depending on accumulation rates.

The observation that GICC05 likely undercounts ice layers during GS-1and H1/GS-2 is quantitatively comparable to previous
results (Adolphi et al., 2018; Muscheler et al., 2014b). Specifically, COSMO highlights that GICC05 counts on average ~7.5% too few annual layers at ~12 kyr b2k, while both CLIM and COSMO consistently indicate that the Greenland timescale likely counts ~10% too few layers in the interval ~15-19 kyr b2k (Fig. 7c). The results from CLIM suggest that within GS-4 and H4/GS-9, GICC05 counts on average ~20% too few annual layers (Fig. 7c). The relative counting error during GS-4 cannot be counter-checked using COSMO, but the latter confirms that across H4/GS-9 GICC05 overcounts annual layers by ~10%.
The larger estimates of the relative counting error in CLIM with respect to COSMO are arguably a result of the low noise-to-signal ratio in the climate records relative to the cosmogenic radionuclide data used for synchronization. The counting error inferred from COSMO over stadials should therefore be deemed as more reliable.

Perhaps a more interesting result is that throughout most of LGM/GS-3 there is an increasing tendency to count too many years in the GICC05 stratigraphy (Fig. 6). The overcount starts at ~23 kyr b2k and reaches a maximum rate of change at ~26
kyr b2k, when the GICC05 timescale counts on average ~25% (CLIM) and ~15% (COSMO) too many layers (Fig. 7c). The LGM/GS-3 interval is the coldest section of the last glacial period (Fig. 7a-b) and an overcount of annual layers is seemingly at odds with the general assumption that during stadials fewer years have been detected, i.e. when low accumulation rates and thinner ice layers make the identification of annual layers more difficult (Rasmussen et al., 2006; Svensson et al., 2008, 2006). However, a bias towards counting too many ice years during LGM/GS-3 is not unexpected as there are some weak indications
that additional annual layers have been counted in other cold sections of the GICC05 stratigraphy (Andersen et al., 2006; Rasmussen et al., 2006; Svensson et al., 2006).

This finding requires further consideration. During LGM/GS-3, the relative counting error inferred from CLIM and COSMO both map reasonably well onto the dust concentration profile in Greenland ice cores (Fig. 7b-c). This interval features the two most distinct and pronounced dust peaks of the last glacial period, in which dust levels increase by a factor of 3 in NGRIP ice
cores (Ruth et al., 2003). Since high dust content in the ice is notoriously liable to complicate the annual-layer counting in a number of ways, this correspondence suggests a possible impact of dust deposition on the identification of the annual layers.

The layer counting in the coldest climatic events of the GICC05 stratigraphy relies mostly on the visual identification of annual variations in two parameters over the NGRIP ice cores. Since the chemical records do not resolve the thin stadial layers, counting is constrained using the high-resolution visual stratigraphy (VS) grey-scale refraction profile (Svensson et al., 2005)
and the electrical conductivity measurement (ECM) on the solid ice (Dahl-Jensen et al., 2002; Hammer, 1980). The VS profile represents the depositional history at NGRIP. Inspections of the VS data throughout the glacial period highlights a strong





correlation between the frequency of visible layers and dust concentration, suggesting that the intensity (i.e. the grey value) of each layer is related to its impurity content representing an individual dust depositional event (Svensson et al., 2005). This may results in the VS record to contain multiple visible large layers per year, which can complicate the counting and lead to a
misinterpretation of the actual annual signal. On the other hand, the ECM is strongly dominated by variations in dust (e.g. Taylor et al., 1997). The ECM profile is attenuated in sections with high concentrations of dust due to the increased alkalinity, thus subduing the annual cycle in the ECM signal and making the resolution of this parameter marginal for the identification of annual layers (Andersen et al., 2006; Rasmussen et al., 2008). For these reasons, greater dust deposition rates may limit the use of the VS and ECM data for direct counting of annual layers.

Moreover, accurate counting over the cold sections that feature multiple depositional events depends on the untested assumption that clusters of peaks in the VS and ECM profiles reflect seasonal variations in dust deposition resembling those observed in the shallower parts of the NGRIP ice core. Modern dust emissions from Asian deserts peak in the Northern Hemisphere spring. This peak is generally associated with enhanced flux of dust to the ice (Beer et al., 1991; Bory et al., 2002; Whitlow et al., 1992) –a signature consistent with that of the warmest sections of the GICC05 stratigraphy and coinciding with
layers of high refraction in the VS record (Ram and Koenig, 1997; Rasmussen et al., 2006; Svensson et al., 2008). However, an altered  atmospheric circulation and precipitation pattern during the LGM may have caused fundamental changes in the seasonality, magnitude, frequency and mode of deposition of dust impurities to the Greenland Ice Sheet.

Model simulations of the dust cycle under glacial climate conditions show a prolongation of the dust-emission season with a two- to three-fold increase in atmospheric emissions and deposition rates in the high northern latitudes during the LGM
compared to modern (Werner et al., 2002). The dominant factor driving the higher dust emission fluxes at the LGM appears to be increased strength and variability of glacial winds over the dust source regions. Evidence from general circulation models (Kageyama and Valdes, 2000; Li and Battisti, 2008; Löfverström et al., 2016; Ullman et al., 2014) and proxy reconstructions (Cheng et al., 2021; Kageyama et al., 2006; Luetscher et al., 2015; Spötl et al., 2021) consistently point towards stronger northern westerlies at the LGM. In particular, GS-3 stands out in the context of the last glacial period as the phase when this
altered flow pattern was most extreme (Fig. 7d), i.e. in conjunction with the maximum extent of the Laurentide Ice Sheet, which caused a strengthening and southward deflection of the westerlies (e.g. Löfverström and Lora, 2017).

Increased emissions and transport to Greenland provide an explanation for the high dust concentrations in the ice observed during LGM/GS-3. However, to justify that GICC05 contains 15-25% too many annual layers during GS-3, additional factors have to be invoked. For example, changes in the seasonality and mode of precipitation can play a key role in increasing the
number of dust depositional events that ultimately control the frequency of sub-annual layers observed in the VS profile. Model simulations suggest that at LGM Greenland experienced a marked reduction in winter and spring precipitation and a shift to a precipitation regime with a pronounced summer maximum –in contrast to the characteristic modern springtime peak (Krinner et al., 1997; Merz et al., 2013; Werner et al., 2002). It has been shown that lower precipitation rates and a shift in seasonality of precipitation inhibit wet deposition of dust during glacial winter and spring. This leads to a substantial increase in the



contribution of dry deposition processes at Summit, which produce dust spikes in ice cores that are less evenly distributed over depth than modern ones (Werner et al., 2002). Dry deposition commonly occurs through gravitational settling and turbulent redistribution of snow to the surface and is thus more conducive to increasing the frequency of annual dust depositional events registered in ice-core records. Hence, the increased seasonality of LGM precipitation and enhanced dry deposition over Greenland may explain the higher frequency of sub-annual layers in the VS signal and the resulting overcount of annual layers

in GICC05 during GS-3. A complete understanding of the physical processes that led to the overcount during GS-3 is however beyond the scope of this work and requires more detailed investigations using climate model experiments of dust transport and deposition.

## 4 Conclusions

The first continuous synchronization between the Greenland ice-core chronology 2005 (GICC05) and the Hulu Cave U-Th

timescale is presented. The synchronization was established using an automated alignment algorithm for Bayesian inversion of the maximum counting error (MCE) of GICC05. The algorithm quantifies the probability of alignments between Greenland ice-core and Hulu Cave proxy signals, and infers the age difference between the two timescales. The synchronization method considers prior information from the MCE but evaluates possible shifts in the timescale difference that exceed the differential dating uncertainty of GICC05, which are not easily quantifiable using traditional tie-point correlation or wiggle-matching

techniques.

To control the accuracy of the timescale difference, two independent synchronizations were generated: one based on climate proxy records (CLIM) and one independent of climate based on cosmogenic radionuclide records (COSMO). The CLIM and COSMO synchronizations are coherent throughout the last glacial period and improve the average precision of the GICC05-Hulu Cave timescale transfer function by ~40% and ~25% relative to previous estimates, respectively. Based on the assumed

accuracy of the U-Th timescale, the results significantly reduce the absolute dating uncertainty of the Greenland timescale back to 48 kyr b2k and indicate that the MCE is generally a conservative uncertainty measurement.

Yet, the analysis shows that the relationship between the GICC05 and the U-Th timescale is more variable than previously assumed and that the annual-layer counting error of the ice-core chronology is not necessarily correlated over long periods of time. It is found that within a number of Greenland Stadials (GS), GICC05 is either missing or gaining time faster than allowed

by its nominal differential dating uncertainty. The annual-layer count identifies on average 7.5-20% too few ice years within GS-1 (the Younger Dryas period), GS-2 (Heinrich event 1), GS-4, and GS-9 (Heinrich event 4), In contrast, up to 15-25% too many ice years may have been counted within GS-3 (the Last Glacial Maximum, LGM).

The results imply a major shift in the differential counting within the interval ~22-27 kyr b2k, when the difference between the GICC05 and the U-Th timescale drifts from +480 to -40 years. The reason for this marked overcount is attributed to a

misinterpretation of the annual-layer record over GS-3. This is likely due to an increased occurrence of multiple-layer years resulting from a higher frequency of dust depositional events at the LGM in response to changes in seasonality of precipitation





and a greater contribution of dry deposition processes. This is an important point, as a large counting bias within GS-3 may explain why it has been difficult to identify a robust bipolar volcanic match between Greenland and Antarctic ice cores during LGM (Svensson et al., 2020).

This study illustrates the utility of probabilistic inversion methods to infer continuous and objective synchronizations of paleoclimate records with a high degree of precision and accuracy. The high-precision timescale transfer functions presented here set important constraints on the biases that accompany the stratigraphic dating of GICC05 and will facilitate the comparison of ice cores, U-Th-dated and radiocarbon-dated records on a common timeline.

**Data availability.** The climate (CLIM) and climate-independent (COSMO) transfer functions presented in Figure 4-6 are
available as a supplement to this paper.

**Code availability.** All R code used for synchronization analysis is available from the author upon request.

**Competing interests.** The author declares no conflict of interest.

**Acknowledgements.** The author thanks Jai Chowdhry Beeman for helpful comments on an earlier version of this manuscript. This study is a contribution to the INTIMATE (INTegration of Ice-core, MArine, and TErrestrial records) project.



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




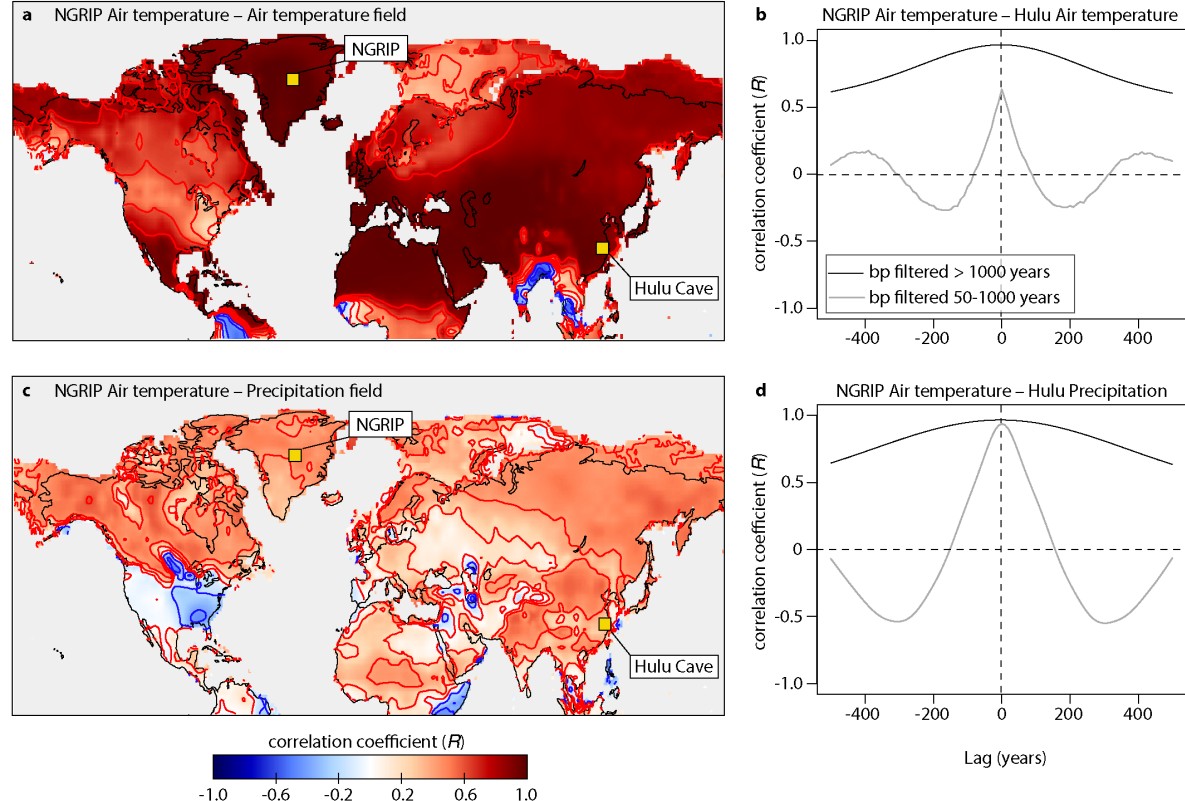

**Figure 1. a.** Instantaneous (lag 0) spatial correlation between mean decadal surface air temperature at the NGRIP site and mean decadal land surface temperatures during the last glacial period (11-60 kyr b2k) as simulated with the HadCM3B-M2.1 coupled general circulation model (Armstrong et al., 2019). The simulation incorporates Dansgaard-Oeschger cycles, Heinrich events, and shorter-term variability, with a

spatial climate fingerprint derived from a last-glacial maximum freshwater hosing experiment applied over the North Atlantic Ocean. The location of NGRIP and Hulu Cave are also shown. **b.** Cross-correlation between simulated NGRIP and Hulu Cave mean decadal air temperature between 11 and 60 kyr ago. The timeseries were bandpass filtered to quantify leads and lags at millennial and shorter timescales, respectively. **c.** Same as (a) but for mean decadal land surface precipitation. **d.** Same as (b) but using Hulu Cave mean decadal precipitation.





**Figure 2.** Proxy-climate data used for the climate synchronization (CLIM) presented in this study and shown on their original timescale. **a.** Mineral-dust derived $Ca^{2+}$ ion concentration record (Erhardt et al., 2019) and deuterium (*d*) excess record from NGRIP (Landais et al., 2018; Steffensen et al., 2008) on the GICC05 timescale (Rasmussen et al., 2006; Svensson et al., 2008). Greenland Interstadials (GI) and partitioning of Greenland Stadial (GS) 2 are indicated. Note the different proxy-data structure during GS-2.1a-b. **b.** High-resolution Hulu Cave $\delta^{18}O$ record (Cheng et al., 2016) on the revised U-Th timescale (Cheng et al., 2018). **c.** Maximum age error (95% confidence level) associated with the U-Th age model (grey), and individual U-Th measurements with their $\pm 1\sigma$ uncertainty (red squares) (Cheng et al., 2018). **d.** Stack of NGRIP $Ca^{2+}$ and Hulu $\delta^{18}O$ records using a technique in which 13 individual events are centred at the midpoint of their abrupt transition, i.e. either DO warming (onset of GIs) or DO cooling (onset of GSs) (note the reverse scale). The events were normalized and





averaged to highlight the shared climatic signal at multidecadal and centennial timescales (>50 year low-pass filtered) and compare the duration of the abrupt DO transitions between Greenland ice cores and Hulu Cave speleothems. Shading reflects the variability across the events used for stacking. The midpoints of the abrupt transitions were identified using a Bayesian change-point analysis method (Erdman and Emerson, 2007).



**Figure 3.** Cosmogenic radionuclide records used for the climate-independent synchronization (COSMO) presented in this study and shown on their original timescale. **a.** Stack of high-resolution Greenlandic and Antarctic ice-core records of ${}^{10}$Be fluxes on the GICC05 timescale. The stack incorporates ${}^{10}$Be data from GRIP (Adolphi et al., 2014; Baumgartner et al., 1997; Wagner et al., 2001; Yiou et al., 1997), NEEM (Zheng et al., 2021), GISP2 (Finkel and Nishiizumi, 1997), NGRIP, EDC, EDML and Vostok (all presented in Raisbeck et al., 2017), and WDC (Muschitiello et al., 2019). Grey bars at the top of the figure indicate the time span of each ${}^{10}$Be record. The Antarctic records were



placed on the GICC05 timescale using the latest bipolar synchronization based on the volcanic tie points presented in (Svensson et al., 2020). **b.** Modelled $^{10}$Be-based $\Delta^{14}$C values with 68% and 95% uncertainty shading (dark grey and grey, respectively) based on individual Monte

Carlo realizations of the stacked $^{10}$Be flux data. **c.** Atmospheric $^{14}$C concentrations based on the Hulu Cave high-resolution data together with their $\pm 1\sigma$ uncertainty (Cheng et al., 2018). **d.** Shortest resolvable bandwidth based on the Nyquist sampling criterion shown here with its 95% probability range derived from the uncertainty associated with the Hulu Cave U-Th chronology. Horizontal dashed lines reflect the bandwidth of the filter applied to the $^{14}$C data to investigate centennial-scale variations in solar activity (150-500 years). Black bars at the bottom of the figure indicate the lower-resolved or analytically uncertain sections of the Hulu Cave $\Delta^{14}$C record that have not been considered

for synchronization.

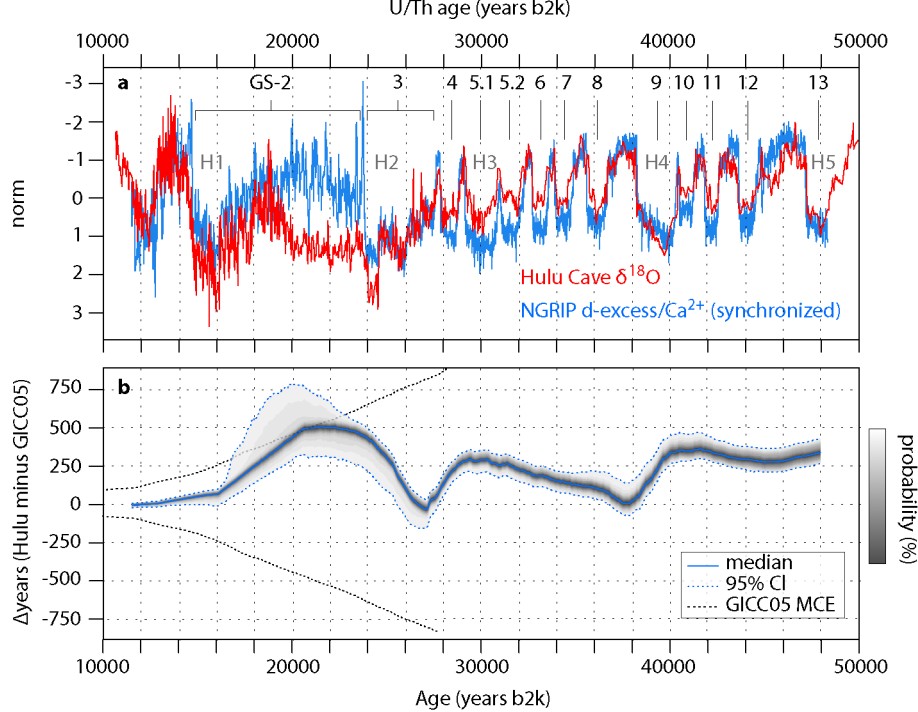

**Figure 4.** Climate synchronization (CLIM) of GICC05 to Hulu Cave records during the last glacial period and resulting timescale transfer function. The synchronization spans the interval 11.5-48 kyr b2k and was derived from stratigraphic alignment of the combined Greenland $d$-excess/$Ca^{2+}$ data to Hulu Cave $\delta^{18}O$. **a.** Synchronized Greenland $d$-excess/$Ca^{2+}$ data on the Hulu Cave U-Th timescale using the posterior median estimate of the MCMC synchronization. The proxy records are presented in normalized units. **b.** Posterior median (blue line) and pointwise 95% credible intervals (shading and blue dashed lines) of the difference $\Delta T$ between the GICC05 and Hulu Cave U-Th timescales. Greenland Stadials (GS) and timing of Heinrich events (H) are indicated at the top.



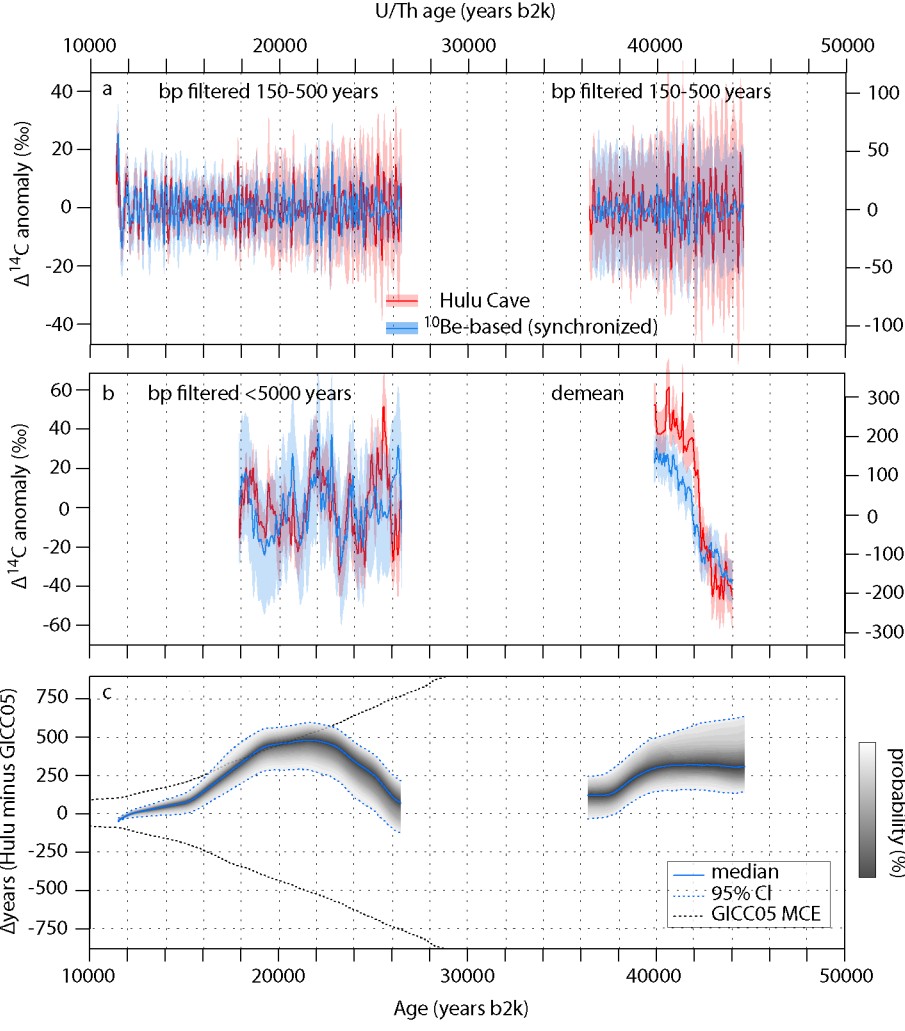

**Figure 5.** Synchronization of GICC05 to Hulu Cave records based on cosmogenic radionuclides data (COSMO) and resulting timescale transfer function. COSMO spans two distinct periods: deglaciation and the Last Glacial Maximum (i.e. 11.5-26.5 kyr b2k), and the Laschamp geomagnetic excursion (i.e. 36.5-45 kyr b2k). The synchronization was obtained by simultaneously aligning centennial- and millennial-scale variations in $\Delta^{14}$C anomalies of the Hulu Cave record and the modelled $\Delta^{14}$C data derived from the ice-core $^{10}$Be stack. **a.** Synchronized centennial (150-500 years band-pass filtered) $\Delta^{14}$C anomalies on the Hulu Cave U-Th timescale using the posterior median estimate of the MCMC synchronization. Shading reflects the $\pm 1\sigma$ uncertainty of individual Hulu Cave $^{14}$C measurements (red) and the modelled $^{10}$Be-based $\Delta^{14}$C values, i.e. conservatively assumed to be $\pm 10$‰ (Adolphi et al., 2018). **b.** Same as (a) but for variations on longer timescales (<5000 year high-pass filtered for the interval 18-26.5 kyr b2k, and demeaned data for the interval 40-44 kyr b2k). Shading as in (a) but assuming $\pm 1\sigma$ uncertainties of $\pm 30$‰ over the period 18-26.5 kyr b2k, and $\pm 50$‰ during 36.5-45 kyr b2k (Adolphi et al., 2018). Note different scaling in (a) and (b) for the two intervals. **c.** Posterior median (blue line) and pointwise 95% credible intervals (shading and blue dashed lines) of the difference $\Delta T$ between the GICC05 and Hulu Cave U-Th timescales.


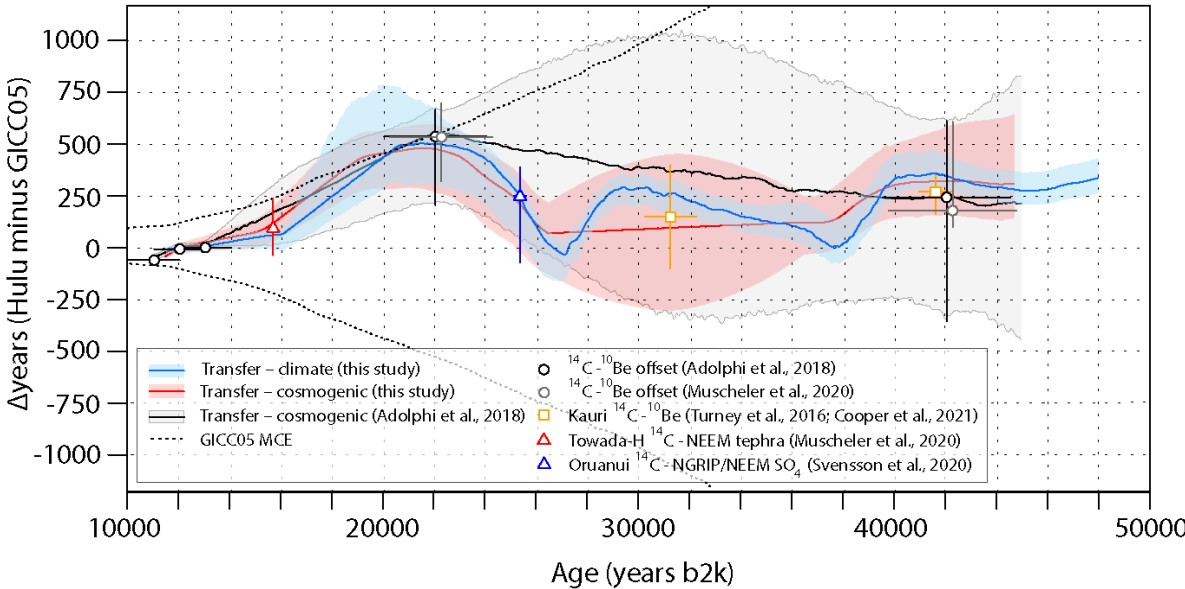


**Figure 6.** Posterior timescale transfer functions based respectively on the climate (CLIM) and the climate-independent (COSMO) MCMC synchronization of the GICC05 to the Hulu Cave U-Th timescale. Positive values indicate that the U-Th timescale is older than GICC05. The transfer functions are presented with their median (thick lines) and pointwise 95% credible intervals (shading). The results are compared to the transfer function presented in Adolphi et al. (2018), which is based on a compilation of U-Th-dated [14]C records, including the low-resolution and less precisely dated Hulu Cave data (Southon et al., 2012). The markers with error bars ($\pm 2\sigma$) show discrete match points inferred from comparison of ice-core [10]Be records and absolutely dated [14]C data, or from [14]C-dated volcanic eruptions identified in Greenland ice cores (Muscheler et al., 2020; Svensson et al., 2020). Note that the gap in the COSMO synchronization (i.e. 26.5-36.5 kyr b2k) has been filled using an autoregressive (AR) process model with similar AR properties as the GICC05 maximum counting error (see section 3.1 for details).



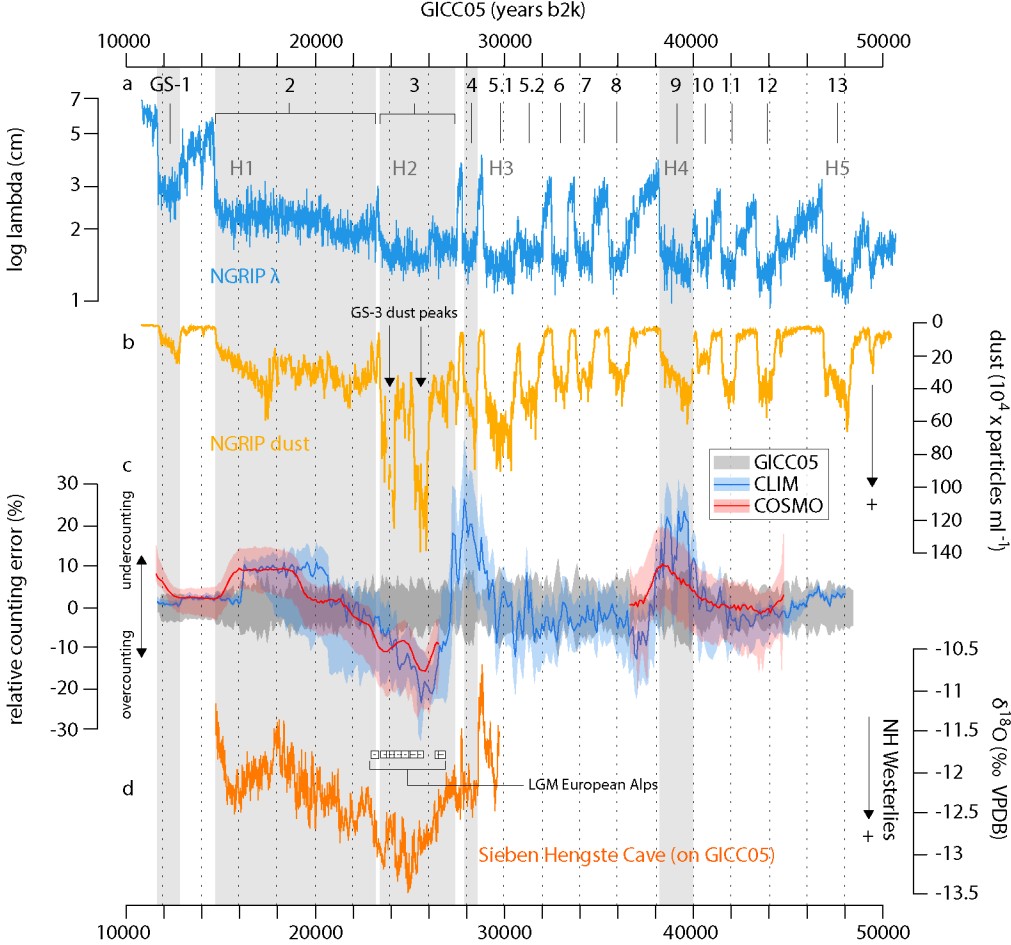


**Figure 7.** Inferred estimates of the relative annual-layer counting error for the GICC05 timescale based on the MCMC synchronizations to Hulu Cave records presented in this study. **a.** NGRIP annual layer thickness (Erhardt et al., 2019) presented with partitioning of Greenland Stadials (GS) and timing of Heinrich events (H). **b.** NGRIP insoluble dust concentration record (Ruth et al., 2003) (note the reverse scale). **c.** Comparison between the maximum relative counting error of the GICC05 (grey shading) and the differential dating uncertainties inferred

from the CLIM (blue) and COSMO (red) synchronization, respectively, presented with their posterior median (thick lines) and pointwise 95% credible intervals (shading). The curves are displayed as 20 year averages and the relative counting error is determined over 100 years. Positive (negative) values indicate an undercount (overcount) of ice layers in Greenland ice cores. Greenland Stadials associated with a relative counting error exceeding that expected by the GICC05 are marked in grey. **d.** U-Th dated climate records of the Last Glacial Maximum (LGM) in the European Alps after synchronization to the GICC05 timescale by applying the CLIM transfer function. U-Th ages

of cryogenic cave carbonates (Spötl et al., 2021) with their ±2σ uncertainty (white squares) indicating the timing of the maximum mountain glacier extent over the European Alps, and δ$^{18}$O values of precipitation from the Sieben Hengste stalagmite record (Luetscher et al., 2015) (orange). The Sieben Hengste record reveals a maximum strengthening and southerly displacement of the westerly winds during GS-3. All records are presented on the GICC05 timescale.