# Peer review of "An improved and continuous synchronization of the Greenland icecore and Hulu Cave U-Th timescales using probabilistic inversion"

_Climate of the Past, 2021_

## Referee Comment (RC2)

Review of Muschitiello 2021, Climate of the Past Discussions by Florian Adolphi

Summary:

Muschitiello presents an assessment of timescale differences between Greenland ice cores and the Hulu Cave speleothems. On one hand, he uses cosmogenic radionuclides (14C & 10Be) and thus, provides an update on Adolphi et al. (2018). On the other hand, he uses climate proxy data (deuterium excess and Ca in the ice cores and d18O in the speleothem) and assumes synchroneity of both signals – an approach that has been used many times in the literature, albeit typically only on specific tie-points, such as the rapid onset of DO-events. Contrary to previous studies, he uses a probabilistic method, which continuously models the ice core chronology, and evaluates the solution by using cost-functions based on the match of cosmogenic radionuclide records (COSMO) and climate records (CLIM).

COSMO essentially confirms the results by Adolphi et al. 2018, but also finds matches between 23-27 kaBP, which results in a shorter period having to be bridged by interpolation to Laschamps. This brings down the stated uncertainty substantially. CLIM agrees with COSMO, but is more continuous. However, CLIM provides more (fast) changes to the timescale difference between Hulu and GICC05, which the author interprets as ice-core layer-counting errors that exceed the stated uncertainties of GICC05.

I appreciate that the probabilistic approach to the synchronization of radionuclides is a step ahead compared to our earlier attempts. However, I do not think that the paper provides sufficient evidence to ensure the robustness of the presented results. Many methodological details and assumptions need explicit testing and explanation and a more critical approach to the data would be advisable.

Major Comments:

14C:

When we did our paper in 2018, we focussed on structures in $\Delta 14C$ that are replicated in more than one archive, and discussed differences when apparent (such as around Laschamps), and still referred to the tie-point around 21ka as "tentative" because the signal to noise is very low. The present study, however, exclusively relies on the Hulu Cave dataset. While I agree that this is the most suitable dataset for this study in terms of resolution (albeit not quite, see below) and measurement uncertainty, I am still doubtful about the signals the author uses for synchronization which may very well be simply noise (see figure R1).

1. The sampling resolution and U/Th-error of the Hulu 14C record is too low to reliably reconstruct the 150-500 year frequency band. The Nyquist wavelength is always at least at the lower bound of this filter or even larger. The author should be aware, that while frequencies lower than the Nyquist frequency may be detected, higher resolution is required to reliably estimate phase and amplitude of the signal needed for a reliable synchronization.
2. The $\Delta 14C$ measurement uncertainty is always substantially larger than the variability (signal) contained in the data in this frequency band (figure R1 middle panel).
3. The $\Delta 14C$-variability in this frequency band is up to an order magnitude larger than what we observe during the Holocene. This is important! The whole paradigm behind the synchronization of radionuclides is, that we are synchronizing production rate changes driven by solar activity or the geomagnetic field. On these frequencies, we should be mainly looking at solar activity changes. So, at face value, this would imply that the Sun was a lot more

variable? 10Be does not seem to support this. This requires a detailed evaluation whether this could be i) due to the carbon cycle, ii) lower geomagnetic field intensity, or iii) simply noise. Given that the increase in variability roughly follows the increase of the measurement uncertainty, I speculate it may be the latter.

4. All these factors combined lead to the fact, that there is no robust phase information in this frequency band. Performing a wavelet analysis of 1,000 random realizations of the Hulu Cave dataset, I obtained a 95% CI of the detected absolute phases that covers nearly 180° - i.e., the full circle. It seems that only at wavelengths >1,500 years, we are seeing phase-signals above noise level.

[Figure]

Figure R1. **Top panel:** Nyquist wavelength (black/grey) and U/Th 2σ dating error (orange). **Middle panel:** Hulu Cave Δ14C measurement uncertainty (black). Green: 2σ-variability within a 2000-year sliding window of the band-pass filtered (150-500 yr) Hulu Cave Δ14C-data. Red: 2σ variability of band-pass filtered Δ14C during the Holocene (IntCal20). **Lower Panel:** absolute phase difference between the lower and upper 95% CI of the phases detected by wavelet analysis of 1,000 random realizations of the Hulu Cave Δ14C record perturbed within its measurement uncertainties.

10Be

Similarly, for the 10Be records I think the information the data can provide should be critically evaluated. Except for the dataset by Raisbeck et al. (2017), only GRIP (whole study period) and WAIS (back to 18.5ka) can resolve the 150-500 year frequency band over extended periods of time. Hence, adding more low-resolution records to the stack will likely only induce biases to the stack. I also don't understand why the author weighs records by resolution? The low-resolution data is not less reliable and should obtain equal weight, since the goal of stacking is to remove depositional/transport-induced signals, where low resolution is quite possibly an advantage.

Regarding the inclusion of the WAIS record into the stack, I wonder how the author treated the fact, that Svensson et al. (2020) provide no tie-points between GICC05 and WDC2014 between 16.5 and 24.5 – a period where COSMO implies large counting errors in GICC05. If this was true, we would need to assume that WDC2014 has the same bias as GICC05, otherwise, one is stacking 2 records that quickly drift apart in their timescales.

So essentially, the only record left to address 10Be high-frequency variability between 16.5 and ~40k is GRIP. I think it would be good to hence, only use this record for the high-frequency part, and use a non-weighted stack of all records for the low-frequency component.

Furthermore, the author suggests that large 10-20% stretches of GICC05 are necessary to fit Hulu 14C. If this is true, than this impacts snow accumulation rates and thus, 10Be fluxes. Hence, before estimating the cost-function, the flux would need to be updated accordingly, and the record filtered again. Generally, when filtering with such a narrow band, the filtering needs to be done on the stretched record, before calculating the cost function, since the frequencies of the record change with stretching.

Along the same lines: The author uses the Svensson et al. synchronization, but AICC12 accumulation-rates for the Antarctic records. But these two are inconsistent. The accumulation rates need to be updated accordingly.

As outlines above, I believe that the 150-500 year frequency band is likely an unreliable target. However, the <5k Band requires an updated analysis of the carbon cycle effects on 14C, since this is not negligible any longer as we are in the same timescale as DO-events and AIMs. In fact, (Cheng et al., 2018) have pointed out that they see climate related variability in the Hulu 14C record. This needs to be addressed.

What drives the synchronization between 22 and 26 kaBP? Neither of the frequency bands look like a good fit between 10Be and 14C?

Have you tested the influence of the dead carbon fraction of Hulu on the phase of signals? We have attempted this in our paper and found the effect non-negligible. Could this be included in the model?

CLIM:

The synchronisation method uses a cost function based on explained variance and root mean square error. Both of these measures imply a linear relation between the two compared variables. For 14C and 10Be this assumption can be assumed to be sufficiently correct if all production rate models and carbon cycle changes are accounted for. But can we assume that deuterium excess or (the logarithm of) Ca are linearly related to Hulu d18O? Even if on the very large scale, they may respond to the same re-organization of the climate system, they respond to very different physical processes, and record different reservoirs and different processes. If this was a good assumption, why do dxs and Ca look differently? And event after synchronization there are many large differences between Hulu d18O and the ice core records especially <25ka BP. Also: it is unclear how the author combined dxs and Ca into one record.

Generally I wonder, how the method evaluates what a "good fit" is. There will obviously always be a best fit, but it may still not be very good. In my opinion a metric for this should be added for both COSMO and CLIM because this may give an indication of the reliability of the synchronization through time.

Last but not least: The author discusses the results of CLIM only in the light of GICC05 counting errors. However, while Corrick et al. (2020) show that the different monsoon regions respond synchronously on average, differences between individual records may be in the order of centuries, likely due to low signal to noise and dating uncertainties. Since the author only uses the Hulu Cave d18O record, it seems premature to only discuss this in the light of GICC05 counting errors, as this may in part originate in the Hulu cave record instead.

Suggestions:

I think the paper should be re-focussed on developing and testing the proposed method more explicitly and thoroughly. The method has the potential to improve the current estimates of the GICC05-Hulu timescale differences, but especially when relying only on one 14C-dataset, more work is needed to show the robustness of the results. In the high-frequency band, the influence of signal to noise needs to be addressed and most importantly: How do we explain the large amplitude of the Δ14C changes? It needs to be shown, that this is most likely production-related to fulfil the premise of the method. For the low-frequency band the effect of carbon cycle changes is likely important. It would be informative if the author showed the influence of the different targets (high-res, low-res, MCE) on the results separately and how this would change if one defined a less narrow frequency band. For CLIM, the premise of the method (linear relation) should be critically evaluated and whether this is a good assumption even outside DO-type variability. For both approaches, it would be good to have a metric of the quality of the fit through time. I agree with reviewer one, that it would be more convenient to invert accumulation instead of age and use this information to update the fluxes before calculating the cost-function.

References:

Adolphi, F., Bronk Ramsey, C., Erhardt, T., Lawrence Edwards, R., Cheng, H., Turney, C.S.M., Cooper, A., Svensson, A., Rasmussen, S.O., Fischer, H., Muscheler, R., 2018. Connecting the Greenland ice-core and U/Th timescales via cosmogenic radionuclides: Testing the synchroneity of Dansgaard-Oeschger events. Clim. Past 14, 1755–1781. https://doi.org/10.5194/cp-14-1755-2018

Cheng, H., Edwards, R.L., Southon, J., Matsumoto, K., Feinberg, J.M., Sinha, A., Zhou, W., Li, H., Li, X., Xu, Y., Chen, S., Tan, M., Wang, Q., Wang, Y., Ning, Y., 2018. Atmospheric 14C/12C changes during the last glacial period from Hulu Cave. Science (80-. ). 362, 1293–1297. https://doi.org/10.1126/science.aau0747

Corrick, E.C., Drysdale, R.N., Hellstrom, J.C., Capron, E., Rasmussen, S.O., Zhang, X., Fleitmann, D., Couchoud, I., Wolff, E., 2020. Synchronous timing of abrupt climate changes during the last glacial period. Science (80-. ). 369, 963 LP – 969. https://doi.org/10.1126/science.aay5538

Raisbeck, G.M., Cauquoin, A., Jouzel, J., Landais, A., Petit, J.R., Lipenkov, V.Y., Beer, J., Synal, H.A., Oerter, H., Johnsen, S.J., Steffensen, J.P., Svensson, A., Yiou, F., 2017. An improved north–south synchronization of ice core records around the 41 kyr 10Be peak. Clim. Past 13, 217–229. https://doi.org/10.5194/cp-13-217-2017

Svensson, A., Dahl-Jensen, D., Steffensen, J.P., Blunier, T., Rasmussen, S.O., Vinther, B.M., Vallelonga, P., Capron, E., Gkinis, V., Cook, E., Kjær, H.A., Muscheler, R., Kipfstuhl, S., Wilhelms, F., Stocker, T.F., Fischer, H., Adolphi, F., Erhardt, T., Sigl, M., Landais, A., Parrenin, F., Buizert, C., McConnell, J.R., Severi, M., Mulvaney, R., Bigler, M., 2020. Bipolar volcanic synchronization of abrupt climate change in Greenland and Antarctic ice cores during the last glacial period. Clim. Past 16, 1565–1580. https://doi.org/10.5194/cp-16-1565-2020

---

## Author Comment (AC1)

(R1 comments in normal typeface; **responses in bold**)

After reviewing this manuscript, I have a mixed feeling about its quality. The introduction is clear and well written. The discussion section is also interesting and well presented. But the method section, with its description of the synchronisation method is in my opinion unclear, and probably contains some mistakes. An effort is therefore needed in my opinion to better describe this method. If I understood correctly, the age transfer function is supposed to be continuous and linear by parts with only 4 segments, which is a very restrictive assumption that should be discussed in greater detail. Moreover, I am personally not convinced that current automated synchro methods can better synchronize than the human brain when the signal-to-noise ratio is low.

**Many thanks for the detailed scrutiny of the methodology section. This is welcome and much appreciated. I agree that there are points that can be improved and I will address all the specific points and typesetting mistakes that the reviewer has brought up. As to the point about potential noise in the data, I will present a new CLIM synchronization that combines multiple speleothem d18O records (please see my responses to R2 and R3).**

**Replies to specific comments of particular concern:**

Eq. (2): First, on the general expression of this cost function. I am personally unfamiliar with this way of adding the R^2 and the RMSD. Where does this come from?

**Thank you for bringing this up. I understand that the combination of R^2 and RMSD can be confusing. Accordingly, the algorithm was modified and the synchronizations were performed again after re-defining the log-likelihood using only the minimum distance between the observed (obs) and simulated (sim) data –i.e. the most commonly employed formulation of the misfit in inverse problem studies:**

$$M(\boldsymbol{X}) = \sum_i \frac{|D_i^{obs} - D_i^{sim}|}{\sigma_i}$$

344: To constrain the ages to be strictly increasing, it would be more convenient to invert positive sedimentation rates.

**Thank you, this is a valid point that should be addressed when dealing with records of cosmogenic radionuclide fluxes. Given R2's comments, the revised manuscript will only focus on the climate synchronization. However I will certainly consider this in future cosmogenic synchronizations.**

Eq. (7): Why are there only 4 segments in the synchronisation, with the last two segments having a slope equal to the average slope (l. 352)? This seems to be a very restrictive way to define a synchronisation. I could not understand if this is a global formulation of the transfer function or only a local formulation. If this is global, it is a very restrictive assumption that should be discussed in greater detail.

**I agree that the formulation is simplistic and I have been open about the fact that the forward model may not provide a realistic representation of the alignment process (see**

**lines 356-357, and 391-395). However, employing random restarts and only a handful of parameters, makes the algorithm computationally much more efficient than other automated methods (e.g. Lin et al., 2014). Furthermore, random restarts minimize the constraints of a 4-segment linear function (see lines 412-429) and effectively allow the algorithm to explore alignment pathways in a fashion that is qualitatively comparable to other more sophisticated routines – capabilities that have been demonstrated in other seminal works (Sessford et al., 2019; Cutmore et al., 2021; West et al., 2021). Although it is hoped that in the future the same synchronizations will be done using more refined automated methods, it is unlikely that the results will be fundamentally different from those presented in this paper. That being said, as recommended by the reviewer, I will discuss strengths and limitations of the forward model in more detail and more upfront in the manuscript. In particular, the importance of random restarts in relaxing the restrictions associated with the piecewise linear function synchronization will be further highlighted.**

**Finally, I would like to point out that I have improved the algorithm and re-performed the synchronizations (see also comments to R2 & 3). Model mixing is now improved using a differential-evolution (DE) MCMC sampler (ter Braak and Vrugt, 2008) – whereby multiple chains are run in parallel in such a way that some can move around the parameter space more easily– while the number of random restarts was increased.**

Section 2.3.5: Nothing is said about the computation time to get the posterior distribution, it would be interesting to know that, since it is generally the Achilles' heel of MCMC methods.

**The runtime of MCMC algorithms is heavily dependent on computer specs. To give the reviewer a ballpark estimate, in the particular case of CLIM using a DE-MCMC sampler in RStudio (v1.3.1093) on a late 2017 MacOS system, the algorithm runs ~1.7k simulations per second.**

**References**
ter Braak, Cajo JF, and Jasper A. Vrugt. "Differential evolution Markov chain with snooker updater and fewer chains." *Statistics and Computing* 18, no. 4 (2008): 435-446.

Cutmore, Anna, Blanca Ausín, Mark Maslin, Timothy Eglinton, David Hodell, Francesco Muschitiello, Laurie Menviel et al. "Abrupt intrinsic and extrinsic responses of southwestern Iberian vegetation to millennial-scale variability over the past 28 ka." *Journal of Quaternary Science* (2021).

Lin, Luan, Deborah Khider, Lorraine E. Lisiecki, and Charles E. Lawrence. "Probabilistic sequence alignment of stratigraphic records." *Paleoceanography* 29, no. 10 (2014): 976-989.

Sessford, E. G., Mari Fjalstad Jensen, Amandine Aline Tisserand, Francesco Muschitiello, T. Dokken, Kerim Hestnes Nisancioglu, and Eystein Jansen. "Consistent fluctuations in intermediate water temperature off the coast of Greenland and Norway during Dansgaard-Oeschger events." *Quaternary Science Reviews* 223 (2019): 105887.

West, Gabriel, Andreas Nilsson, Alexis Geels, Martin Jakobsson, Matthias Moros, Francesco Muschitiello, Christof Pearce, Ian Snowball, and Matt O'Regan. "Late Holocene paleomagnetic secular variation in the Chukchi Sea, Arctic Ocean." (2021).

---

## Author Comment (AC2)

(R2 comments in normal typeface; **responses in bold**)

**I would like to thank the reviewer for the detailed commentary and especially for the time he took to analyse the Hulu 14C data. This is very much appreciated. In the following I will address the main points raised by the reviewer.**

The present study exclusively relies on the Hulu Cave dataset. While I agree that this is the most suitable dataset for this study in terms of resolution and measurement uncertainty, I am still doubtful about the signals the author uses for synchronization which may very well be simply noise.

Hulu Cave 14C
The Δ14C-variability in the 150-500 yr frequency band is up to an order magnitude larger than what we observe during the Holocene. This is important! The whole paradigm behind the synchronization of radionuclides is, that we are synchronizing production rate changes driven by solar activity or the geomagnetic field. On these frequencies, we should be mainly looking at solar activity changes. So, at face value, this would imply that the Sun was a lot more variable? 10Be does not seem to support this. This requires a detailed evaluation whether this could be i) due to the carbon cycle, ii) lower geomagnetic field intensity, or iii) simply noise.

**The points raised by the reviewer are fair and having examined Fig. R1, I acknowledge that beyond deglaciation there is probably no robust phase information to constrain the 14C-10Be synchronization at sub-millennial timescales. The reviewer requested a detailed evaluation of the carbon cycle effects on 14C in the <5ka frequency band, the impact of geomagnetic field intensity, the effect of climate-related variability in the Hulu 14C record, the potential influence of the DCF on the phase of signals, etc. These are valid suggestions. However, considering the significant additional time required to meet these requests and the fact that the results are unlikely to confirm that production rates dominate the multi-centennial frequency band, I believe these additional analyses would better fit the scope of a dedicated paper using ad-hoc carbon cycle sensitivity experiments. Thankfully the main conclusions of the present study do not depend on COSMO, which mainly served as supporting evidence. In light of this consideration and the reviewer's comments I think a more feasible approach is to shift the focus of the paper to the CLIM results and address issues and concerns surrounding the climate synchronization instead.**

**If the editor considered this to be important, I would still be willing to present the COSMO results for selected intervals such as in Adolphi et al. (i.e. deglaciation, 21-24ka). However, preliminary results suggest that they would not significantly improve the estimates presented by Adolphi and colleagues.**

10Be
Regarding the inclusion of the WAIS record into the stack between 16.5 and 24.5 ka, I wonder how the author treated the fact, that Svensson et al. (2020) provide no tie-points between GICC05 and WDC2014 between 16.5 and 24.5.

**This is not an issue as the WDC data only stretch back to ~15ka (see Fig. 3a).**

So essentially, the only record left to address 10Be high-frequency variability between 16.5 and ~40k is GRIP. I think it would be good to hence, only use this record for the high-frequency part, and use a non-weighted stack of all records for the low-frequency component.

**Thank you for the suggestion. I will keep this in mind for future reference.**

Furthermore, the author suggests that large 10-20% stretches of GICC05 are necessary to fit Hulu 14C. If this is true, than this impacts snow accumulation rates and thus, 10Be fluxes. Hence, before estimating the cost-function, the flux would need to be updated accordingly, and the record filtered again.

**This is a valid point also raised by R1. I will take this into account in future cosmogenic synchronizations.**

The synchronisation method uses a cost function based on explained variance and root mean square error. Both of these measures imply a linear relation between the two compared variables. For 14C and 10Be this assumption can be assumed to be sufficiently correct if all production rate models and carbon cycle changes are accounted for. But can we assume that deuterium excess or (the logarithm of) Ca are linearly related to Hulu d18O? Even if on the very large scale, they may respond to the same re-organization of the climate system, they respond to very different physical processes, and record different reservoirs and different processes. If this was a good assumption, why do dxs and Ca look differently? And event after synchronization there are many large differences between Hulu d18O and the ice core records especially <25ka BP. Also: it is unclear how the author combined dxs and Ca into one record. […] For CLIM, the premise of the method (linear relation) should be critically evaluated and whether this is a good assumption even outside DO-type variability.

**The CLIM synchronization has been performed again taking into account a number of considerations raised by all the reviewers (see also replies to R1 and R3). First, I understand that combining d-excess and Ca2+ may be confusing, so I have decided to employ only Ca2+ –the proxy that has been used for the climate synchronization in Adolphi et al. To provide a physical basis for the assumption of a roughly linear –and synchronous– behaviour between hydroclimate shifts at Hulu Cave and Greenland Summit, I have analysed climate model output of the LGM (the greatest concern for R2) from a transient simulation of the last glacial period (Armstrong et al., 2019), a transient simulation of the last deglaciation (Liu et al., 2009), and all the available LGM experiments from the PMIP4-CMIP6 framework. The new analysis confirms the presence of a persistent hydroclimatic covariability at multidecadal and centennial timescales between SE Asia and Greenland during pleniglacial (similarly to the results presented in Fig. 1b & d), which supports the approach used for the climate synchronization at timescales shorter than DO variability. These new findings will be included in the revised version of the manuscript and the linear relationship assumption will be discussed in more detail.**

Generally I wonder, how the method evaluates what a "good fit" is. There will obviously always be a best fit, but it may still not be very good. In my opinion a metric for this should be added for both COSMO and CLIM because this may give an indication of the reliability of the synchronization through time. […] It would be good to have a metric of the quality of the fit through time.

**A similar point was raised by R3 and it comes as a surprise. There seems to be a fundamental misunderstanding here as Bayesian results do not require validation via deterministic metrics –something inconsistent with the framework for fitting inversion problem models and bearing no purpose in the specialised literature. Within the probabilistic approach taken in this study, it is not informative to take on the null-hypothesis framework. One could tackle the synchronization problem with the frequentist approach but that would require determining the alignment *a priori* (e.g. using several tie points) rather than inferring it *a posteriori*, which is the main motif of this research. Estimating the significance of the correlation (or any other metric) to evaluate the 'goodness' of the Bayesian fit between the Hulu and ice core data is in general not a good idea and overall incoherent with the probabilistic approach employed here.**

Last but not least: The author discusses the results of CLIM only in the light of GICC05 counting errors. However, while Corrick et al. (2020) show that the different monsoon regions respond synchronously on average, differences between individual records may be in the order of centuries, likely due to low signal to noise and dating uncertainties. Since the author only uses the Hulu Cave d18O record, it seems premature to only discuss this in the light of GICC05 counting errors, as this may in part originate in the Hulu cave record instead.

**This is a fair comment that has also been raised by R3. I have taken these points on board and re-estimated the climate synchronization accordingly. To minimize the noise due to local environmental factors, account for dating errors, and altogether better represent the large-scale hydroclimatic imprint of the monsoon system, I have applied a Monte Carlo Empirical Orthogonal Function (MCEOF) (e.g. Anchukaitis and Tierney, 2013) approach that integrates all the available speleothem d18O records from SE Asia spanning the last 50ka. I utilized 17 d18O timeseries that were used to define the Asian Summer Monsoon region in Corrick et al. (2020). The compilation includes the U-Th age determinations underlying the speleothem chronologies.**

**The method uses iterative age modelling of the available U-Th ages and eigen-decomposition of the d18O records to isolate the common d18O patter and estimate uncertainties. By randomly resampling the age uncertainties of each speleothem record, I generated a 1,000 member ensemble of the first leading mode of the MCEOF analysis (EOF1), which is dominated by the characteristic regional monsoonal signal of the last glacial period (note that each MC iteration was set such that the sign of the EOF1 is consistent across the ensemble). The Monte Carlo approach results in some temporal smoothing of the EOF1 but does not affect the fidelity of millennial-scale trends or shorter features. The CLIM synchronization was then performed in a way such that for each MCMC iteration the algorithm employs as input NGRIP Ca2+ and as target a randomly resampled (with replacement) EOF1 from the 1,000 member ensemble of the MCEOF analysis described above. It should be noted that the speleothem compilation used here includes an updated version of the Hulu Cave data based on the newly published U-Th estimates presented in Cheng et al. (2021), which significantly improve the resolution and dating precision of the cave record across Heinrich Stadial 4 and bring the d18O data more in line with the trajectories observed in other regional records.**

**Altogether, the MCEOF approach and the updated Hulu Cave chronology improve the robustness of the CLIM synchronization and provide a more representative estimate of the alignment uncertainties.**

**References**

Anchukaitis, Kevin J., and Jessica E. Tierney. "Identifying coherent spatiotemporal modes in time-uncertain proxy paleoclimate records." *Climate dynamics* 41, no. 5-6 (2013): 1291-1306.

Armstrong, Edward, Peter O. Hopcroft, and Paul J. Valdes. "A simulated Northern Hemisphere terrestrial climate dataset for the past 60,000 years." *Scientific data* 6, no. 1 (2019): 1-16.

Cheng, Hai, Yao Xu, Xiyu Dong, Jingyao Zhao, Hanying Li, Jonathan Baker, Ashish Sinha et al. "Onset and termination of Heinrich Stadial 4 and the underlying climate dynamics." *Communications Earth & Environment* 2, no. 1 (2021): 1-11.

Corrick, Ellen C., Russell N. Drysdale, John C. Hellstrom, Emilie Capron, Sune Olander Rasmussen, Xu Zhang, Dominik Fleitmann, Isabelle Couchoud, and Eric Wolff. "Synchronous timing of abrupt climate changes during the last glacial period." *Science* 369, no. 6506 (2020): 963-969.

Liu, Zhengyu, B. L. Otto-Bliesner, Feng He, E. C. Brady, Robert Tomas, P. U. Clark, A. E. Carlson et al. "Transient simulation of last deglaciation with a new mechanism for Bølling-Allerød warming." *science* 325, no. 5938 (2009): 310-314.

---

## Author Comment (AC3)

(R3 comments in normal typeface; **responses in bold**)

1) Greenland and Hulu Cave data covariation.

The study is based on correlation of NGRIP Deuterium excess data with Hulu d18O. The reason is explained in line 157-160 and may be true for the response to the H events (as it is for D-O events), but it is a critical and very, very daring assumption that Hulu d18O and NGRIP D excess trace the same smaller-scale climatic changes. I do not think that this has been demonstrated previously, and the manuscript does not provide compelling evidence that the smaller-scale features correlate significantly. As I understand, Figure 1 does not separate the hosed D-O-scale/H-scale variability from smaller-scale variability.

**Thank you for bringing this up. Fig. 1b & d do already show that climate/hydroclimate changes at NGRIP and Hulu Cave are linearly correlated and synchronous at timescales shorter than DO variability. I will present this finding more clearly in the main text.**

I think the manuscript will either have to demonstrate with statistical back up that the Hulu d18O signal correlates significantly with at least one of the Greenland records (e.g. d18O, Calcium, or D excess) also over smaller-scale (and preferably non-forced) changes, or refrain from presenting a match of the records across these rather long periods which do not have D-O- or H-scale variability. That would challenge the concept of a "continuous" transfer function.

**Notwithstanding the climate model results presented in Fig. 1b & d (see my previous comment), I agree that this issue should be addressed more rigorously. As also discussed in my replies to R2, I will address this concern in the revision. To demonstrate a physical relationship between hydroclimatic changes in the Asian Summer Monsoon region and Greenland Summit at timescale shorter than DO variability, I will present a novel analysis of climate model output from transient simulations of the last glacial period (Armstrong et al., 2019) and deglaciation (Liu et al., 2009), and from PMIP4-CMIP6 LGM experiments. All simulations reveal the presence of a persistent hydroclimatic covariability at multidecadal and centennial timescales between SE Asia and Greenland, which justifies the alignment approach used here –in particular for the interval 18-24ka.**

2) Speleo dating uncertainty

The uncertainties of individual U/Th age determinations are small, but as demonstrated e.g. by the speleothem age-modelling work of Corrick et al., 2020, different but realistic assumptions about growth rates, interpolation methods, purity of samples etc. can lead to differences in ages at a certain speleothem depth that are larger than the raw U/Th age uncertainties (sometimes several times those). Especially at climatic transitions which are not located close to a U/Th-dated sample, this can lead to systematic dating offsets of D-O event onsets. If taken at face value, this forces the duration of the stadials and interstadials to change very significantly and way beyond what it compatible with the constraints from ice-core annual-layer counting (which is exactly what is seen here, described as an ice-core annual-layer-counting bias, line 139-141). This is why Buizert et al., 2014, stretched GICC05 by 1.0063 to fit the Hulu constraints ON AVERAGE and not on a transition-to-transition basis. This can likely be done better with Cheng 2016 data, but both the true age uncertainties

from all sources (and not only the raw U/Th age uncertainties) and uncertainty due to that the D-O onset are not always similar between records must be included (and it is not clear if/how this is presently done). I believe that the current manuscript overemphasizes how tightly this one particular record (the Cheng/Edwards data) with its implicit assumptions about growth model, sample purity etc. can properly constrain individual D-O onset ages with realistic uncertainties, and that this introduces unrealistic stretching/compression of the ice-core time scale. Another way to address this problem would be to use data from other speleothems (e.g. the data from Corrick et al., 2020), and investigate if the results are reproducible under other assumptions.

**Thank you for your suggestion –using multiple speleothem records is a good idea. I agree that dating uncertainties should be accounted for more explicitly in the manuscript. I have taken on board the Reviewer's suggestion and leveraged information from the speleothem compilation presented by Corrick and colleagues. As also discussed in my response to R2, I have devised a Monte Carlo Empirical Orthogonal Function (MCEOF) (e.g. Anchukaitis and Tierney, 2013) to extract the common large-scale d18O signature across a dataset of 17 speleothems from the Asian Summer Monsoon region that accounts for U-Th age uncertainties at each site.**

**The MCEOF method uses iterative age modelling of the available U-Th ages and eigen-decomposition of the d18O records to isolate the common d18O patter and estimate uncertainties. By creating ensembles of possible age models for each speleothem record, I generated a 1,000 member ensemble of the first leading mode of the MCEOF analysis (EOF1), whereby each EOF1 realization is dominated by the characteristic regional monsoonal pattern of the last glacial period. The CLIM synchronization was then performed by employing for each MCMC iteration a randomly resampled (with replacement) EOF1 realization from the 1,000 member ensemble, i.e. used as target for the alignment procedure (Fig. A). Note that the speleothem compilation additionally includes an updated version of the Hulu Cave data based on the newly published U-Th ages presented in Cheng et al. (2021), which significantly improve the resolution and dating precision of the cave record across Heinrich Stadial 4.**

**The MCEOF approach will improve the robustness of the CLIM synchronization and provide a more representative estimate of the alignment uncertainties. Furthermore, the new CLIM results bring about new implications (Fig. B):**

1. **Unlike the results based on Hulu data only, there is an evident systematic ice-layer counting bias of GICC05 that is precisely centred around Heinrich Stadials (i.e. undercounting during H1, H3, H4 and H5, and overcounting during H2);**
2. **The inferred GICC05 relative counting error over these climatic events is more conservative than that presented in the previous version of the manuscript (i.e. within 10-15%);**
1. **The overcounting bias during GS-3 does not exceed 10%.**

[Figure]

*Fig. A – Preliminary CLIM synchronization of GICC05 to the U-Th timescale using the first leading mode from a Monte Carlo EOF procedure based on 17 speleothem d18O timeseries from the Asian Summer Monsoon region (Corrick et al., 2020). The NGRIP Ca2+ data are presented on the U-Th timescale using the posterior median estimate of the MCMC synchronization.*

[Figure]

*Fig. B – Inferred estimates of the relative annual-layer counting error for the GICC05 timescale based on the preliminary synchronization shown in Fig. A. Shading reflect pointwise 95% credible intervals. NGRIP Ca2+ data are also presented (red line) for reference. Note the systematic ice-layer undercounting during H1, H3, H4 and H5. The counting error exceeds the confidence levels predicted by the GICC05 chronology by up to ~5%.*

3) Continuity

The method rests on an assumption that the records can be matched continuously, i.e. that there is robustly correlatable information everywhere in the record. There is always a best match between records being correlated, but the method seems not to address whether "best" is "good enough". It thus becomes impossible for the reader to figure out in which sections the correlation is statistically significant and where there is nothing but noise (or local climate variability etc.) resulting in a transfer function that essentially just bridges between sections with statistically significant correlation.

**I think there is a fundamental misunderstanding here. In a Bayesian context, the term "best fit" is at odds with the classical perspective from inferential statistics rooted in significance testing, hypothesis testing and confidence intervals (in fact, note that I generally refer to "optimal" rather than "best" synchronization). In Bayesian statistics it refers to the best possible fit given some prior estimates of the model parameters (in our case, the parameters defining the synchronization model), and the posterior mean (or median) should be taken as a measure of "adequacy" rather than as a deterministic quantity. There is a fundamental difference in philosophy between frequentist and Bayesian statistics. In the frequentist framework, the model parameters are unknown but deterministic quantities –meaning that the goal is to determine the range of values for the parameters that are supported by the data (i.e. the confidence interval). Bayesian statistics on the other hand is concerned with predicting the data given an estimate of the model parameters –regardless of whether these estimates are meaningful. When the parameters are viewed as deterministic quantities –as in frequentist statistics– it is nonsensical to talk about their probability distributions. Analogously, when parameters are viewed as probability distributions –as in Bayesian statistics– it is nonsensical to talk about their deterministic quantities and statistical significance. For these reasons, Bayesian results do not require validation via deterministic metrics (see also my reply to R2), and estimating the significance of the correlation between the speleothem and ice core data is not informative and overall incoherent with the probabilistic framework adopted here.**

Specific comments

**I will address all the specific points raised by the reviewer following their suggestions.**

Line 287-293 and section 2.3.4: It seems like the method allows that sections of GICC05 are stretched/compressed to fit the assumed perfect Hulu time scale. If needed, sections close to each other can be modified in opposite ways (even though the stretching is smoothed as described in 2.3.3). This seems physically implausible given the nature of the GICC05 counting process: There are likely biases in GICC05, but these are not likely to change abruptly on short time scales (except between interstadials and stadials) because neither the data basis nor the counting method changed quickly. This is mentioned in line 391-395, but I think the results are not at all "approximating the layer-counting structure of the GICC05 timescale" but in stark contrast to the layer counting procedure.

**As discussed in my reply to R1 and openly stated in the main text (see lines 356-357, and 391-395), the synchronization model relies on a simplistic formulation that is not meant to reproduce the complexity of the ice-layer counting and its uncertainty structure. That being said, I concede that interpreting GICC05 counting errors relative to only the**

**Hulu Cave d18O record can be misleading and that some of the features presented in Fig. 7c may be affected by noise and dating uncertainties in the Hulu data set. I think this issue will be mitigated by deploying the MCEOF methodology described above, i.e. following the reviewer's suggestion on combining multiple speleothem d18O records.**

This is also what is seen on Figure 7c: The results indicate that at 26-28 ka, the GICC05 counting bias changes from 20-30% in one direction to 20-30% in the opposite direction. A similar slightly smaller feature is seen around 38 ka. These biases do not correlate with climate: e.g., the phases of largest overcounting happen during a stadial dust peak (26 ka) and during a low-dust interstadial (38 ka). If these features are real and unexplained, there is really no reason to trust GICC05 anywhere in the glacial. Extraordinary claims require extraordinary evidence, and I do not think that the manuscript provides sufficient evidence that these features are real, i.e. cannot at least to a large extent be attributed to weaknesses in the assumptions of Greenland - Hulu Cave climate correlation and underestimation of the total uncertainty of the Hulu Cave record, or alternatively, that the synchronization method does not produce statistically significant results in these sections.

**As the reviewer suggested, integrating multiple speleothem records within the alignment procedure should improve the robustness of the CLIM results and their implications on the GICC05 counting bias. A preliminary synchronization using the MCEOF method (Fig. A & B) indicates that:**
  1. **the relative counting biases are much smaller than initially suggested, i.e. generally they are within 5% larger than the nominal GICC05 uncertainty;**
  2. **the swings from undercounting to overcounting (and vice versa) are more in phase with abrupt climate shifts, especially during H1, H3, H4 and H5.**

**References**

Anchukaitis, Kevin J., and Jessica E. Tierney. "Identifying coherent spatiotemporal modes in time-uncertain proxy paleoclimate records." *Climate dynamics* 41, no. 5-6 (2013): 1291-1306.

Armstrong, Edward, Peter O. Hopcroft, and Paul J. Valdes. "A simulated Northern Hemisphere terrestrial climate dataset for the past 60,000 years." *Scientific data* 6, no. 1 (2019): 1-16.

Cheng, Hai, Yao Xu, Xiyu Dong, Jingyao Zhao, Hanying Li, Jonathan Baker, Ashish Sinha et al. "Onset and termination of Heinrich Stadial 4 and the underlying climate dynamics." *Communications Earth & Environment* 2, no. 1 (2021): 1-11.